# DIFF-IN: DATA INFLUENCE ESTIMATION WITH DIFFERENTIAL APPROXIMATION

## ABSTRACT

In this paper, we introduce a new formulation to approximate a sample's influence by accumulating the differences in influence between consecutive learning steps, which we term Diff-In. Specifically, we formulate the sample-wise influence as the cumulative sum of its changes/differences across successive training iterations. By employing second-order approximations, we approximate these difference terms with high accuracy while eliminating the need for model convexity required by existing methods. Despite being a second-order method, Diff-In maintains computational complexity comparable to that of first-order methods and remains scalable. This efficiency is achieved by computing the product of the Hessian and gradient, which can be efficiently approximated using finite differences of first-order gradients. We assess the approximation accuracy of Diff-In both theoretically and empirically. Our theoretical analysis demonstrates that Diff-In achieves significantly lower approximation error compared to existing influence estimators. Extensive experiments further confirm its superior performance across multiple benchmark datasets in three data-centric tasks: data cleaning, data deletion, and coreset selection. Notably, our experiments on data pruning for large-scale vision-language pre-training show that Diff-In can scale to millions of data points and outperforms strong baselines.

## 1 INTRODUCTION

Data is a driving force behind recent advancements in various fields (Tom Brown et al., 2020; Alexander Kirillov et al., 2023; Robin Rombach et al., 2021), as it directly influences the behavior of learned models, including their performance and inherent biases (Kwon et al., 2023). This highlights the need for a quantitative understanding of how individual data samples affect model learning, which is essential for enhancing both model performance (Yang et al., 2023; Xia et al., 2024) and interpretability (Grosse et al., 2023; Dai & Gifford, 2023).

To address this, influence functions have been introduced (Cook & Weisberg, 1982; Cook, 1986) to study how a specific sample $z$ affects model parameters and loss values:

$$\begin{aligned}
\mathcal{I}_\theta(z) &= \theta^*_{-z} - \theta^*, &&\text{(Influence on parameters)} \\
\mathcal{I}(z, \mathbf{V}) &= \mathcal{L}(\mathbf{V}, \theta^*_{-z}) - \mathcal{L}(\mathbf{V}, \theta^*). &&\text{(Influence on loss)}
\end{aligned} \tag{1}$$

Here, $\theta^*$ represents the learned parameters obtained by optimizing the empirical loss $\mathcal{L}$ on the full training set, while $\theta^*_{-z}$ refers to the parameters learned after excluding the sample $z$. The influence on parameters, $\mathcal{I}_\theta(z)$, also known as Cook's distance (Cook & Weisberg, 1982; Cook, 1986), measures the extent to which the optimized model parameters would change if the sample $z$ is removed from the training dataset. Similarly, the influence on loss, $\mathcal{I}(z, \mathbf{V})$, examines how the model's loss or performance on an evaluation set $\mathbf{V}$ is affected when the sample $z$ is excluded from the training set.

To measure the influence of a sample $z$, a straightforward yet optimal approach would be to remove $z$ from the training dataset and retrain the model to obtain the optimized parameters $\theta^*_{-z}$ – a process known as leave-one-out (LOO) training. However, the retraining is computationally expensive and often impractical. To overcome this limitation, Koh and Liang (Koh & Liang, 2017), building on the formulation of influence functions (Cook & Weisberg, 1982), introduced approximations that

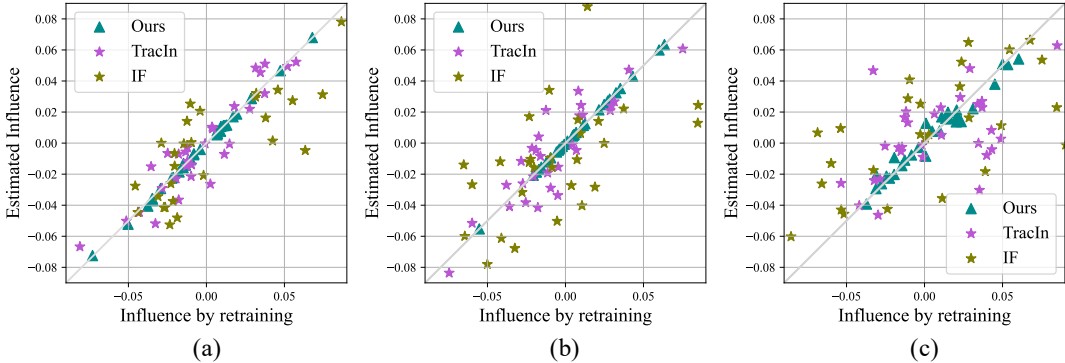

Figure 1: Approximation accuracy comparison by comparing the estimated influence values with actual influence values obtained through brute-force retraining on the 30 most influential data points. There are three model and dataset settings, (a) ResNet-18 on CIFAR-10, (b) ResNet-101 on CIFAR-10, and (c) ResNet-18 on ImageNet-1K. The more accurate a method is, the closer its corresponding scatters will be concentrated near the diagonal. Our approach demonstrates consistent advantages as the model size grows (a vs. b) and as the dataset complexity increases (a vs. c).

estimate a sample's influence without the need for retraining. The key idea is to derive a quadratic approximation of the empirical risk around the stationary point $\theta^*$ and obtain the sample's influence by upweighting it by an infinitesimal amount; see Sec. 2.1 for details. Subsequent studies (Basu et al., 2020; Koh et al., 2019; Basu et al., 2021; Grosse et al., 2023; Kwon et al., 2023; Ko et al., 2024) have expanded on this approach, improving both its efficiency and precision. However, despite these advancements, the accuracy of these methods relies on convexity (Koh & Liang, 2017) of the model – conditions that are rarely satisfied in practice, especially in large models. As a result, these approximations can be inaccurate, as demonstrated in Figure 1 and Figure 3, where the sample's influence (olive stars) deviates significantly from the LOO results, showing a low correlation. This limitation also impacts their performance in practice (see Sec. 5).

Another notable approach, TracIn (Pruthi et al., 2020), offers a heuristic method for approximating a sample $z$'s influence on loss values requiring only first-order gradient computations. This method accumulates the sample's impact on the validation set loss across various training iterations through first-order approximations of the loss. Although TracIn is more computationally efficient, its approximation diverges significantly from the objective defined in Eq. 1, limiting its accuracy (see Figure 1 (purple stars)). Additionally, it cannot be used to estimate a sample's influence on model parameters.

In this paper, we introduce a new perspective on influence estimation by examining its temporal differences, termed Diff-In. The core idea of Diff-In is to represent influence as the cumulative sum of its differences between successive training steps (see Eq. 4). Although simple, this formulation allows us to apply a second-order approximation to each difference term without relying on convexity assumptions (Koh & Liang, 2017) or altering the approximation target (Pruthi et al., 2020), thereby enhancing accuracy (see Figure 1 green triangle). This improvement is demonstrated both theoretically (Sec. 4) and empirically (Sec. 5.4). Moreover, although Diff-In employs a second-order approximation, it does not significantly increase computational complexity compared to existing methods (Koh & Liang, 2017; Pruthi et al., 2020) (see Sec. 5.4). Instead of directly computing second-order derivatives, Diff-In calculates the product of the Hessian and the gradient (Pearlmutter, 1994). This is done using finite differences on the gradient, as shown in Eq. 6, requiring only gradient computations and maintaining a computational complexity comparable to that of first-order methods (Pruthi et al., 2020).

We conduct extensive experiments on various data-centric tasks, including coreset selection (Sorscher et al., 2022), data cleaning (Pruthi et al., 2020), and data deletion (Fu et al., 2022), to evaluate the effectiveness of Diff-In. The results demonstrate that Diff-In consistently outperforms previous methods (Koh & Liang, 2017; Pruthi et al., 2020; Kwon et al., 2023) across all benchmarked tasks and datasets, delivering leading performance in most evaluated scenarios. Notably, Diff-In outperforms all baselines by more than 9.0% in the data cleaning task and achieves a stable performance improvement of over 2.0% in the data deletion task. It also surpasses the widely-used CLIP-score (Alec Radford

et al., 2021) by 1-2% while maintaining a similar time cost in coreset selection for vision-language tasks. Additionally, our approach is versatile, effectively solving all three tasks, while some of the compared methods (Xia et al., 2023; Pruthi et al., 2020; Kim et al., 2024) are tailored to specific tasks and cannot be applied across different data-centric tasks.

## 2 PRELIMINARIES

We have compiled all the notations in Table 5. Let $\mathbf{D} = \{z_0, ..., z_{N-1}\}$ denote the training set, where the number of all samples $|\mathbf{D}| = N$. $\mathbf{D}/z$ is the dataset excluding a sample $z$. For a deep network parameterized by $\theta \in \Theta$, we use $\ell(z, \theta)$ as the loss on a sample $z$ and $\mathcal{L}(:, \theta)$ as the averaged loss over a set of data, where $\mathcal{L}(\mathbf{D}, \theta) = \frac{1}{N} \sum_i \ell(z_i, \theta)$. Please check Eq.1 for the definition of the sample's influence.

### 2.1 REVISITING INFLUENCE ESTIMATION

**Koh and Liang's approach and its subsequent developments.** To mitigate the high cost of brute-force leave-one-out (LOO) retraining, various methods have been proposed to estimate influence (Koh & Liang, 2017; Grosse et al., 2023; Kwon et al., 2023; Pruthi et al., 2020; Basu et al., 2020). A notable example is the approach introduced by Koh and Liang (Koh & Liang, 2017), which calculates the change in model parameters when a sample $z$ is up-weighted by a small amount $\epsilon$. Specifically, the optimal parameters $\theta^*_{\epsilon,z}$, resulting from up-weighting the sample $z$ by $\epsilon$, are formulated as: $\theta^*_{\epsilon,z} = \arg\min_{\theta \in \Theta} \frac{1}{n} \sum_{i=1}^n \ell(z_i, \theta) + \epsilon\ell(z, \theta)$.

Then, according to Cook & Weisberg (1982), by applying a quadratic approximation to the empirical risk around $\theta^*$, the influence of up-weighting $z$ on the parameters $\theta_{\epsilon=0,z} = \theta$ by $\epsilon$ is given by:

$$\mathcal{I}_{\text{up,params}}(z) = \frac{d\theta^*_{\epsilon,z}}{d\epsilon}\bigg|_{\epsilon=0} = -H_{\theta*}^{-1} \nabla_\theta \ell(z, \theta^*), \tag{2}$$

where $H_{\theta*} = \frac{1}{n} \sum_{i=1}^n \nabla_\theta^2 \ell(z_i, \theta^*)$ is the Hessian and is positive definite (PD) by assumption. Since removing a point $z$ is the same as up-weighting it by $\epsilon = -\frac{1}{n}$, one can then approximate the parameter by computing $\mathcal{I}_\theta(z) = \theta^*_{-z} - \theta^* \approx -\frac{1}{n}\mathcal{I}_{\text{up,params}}(z)$.

With $\mathcal{I}_\theta(z)$, the influence on the loss over the validation set $\mathbf{V}$ can be estimated as: $\mathcal{I}(z, \mathbf{V}) = \langle \nabla\mathcal{L}(\mathbf{V}, \theta^*), \mathcal{I}_\theta(z) \rangle$. While these methods represent significant progress, they rely on the assumption that the empirical risk is strongly convex with respect to the parameters—an assumption that is rarely satisfied in practice (Choromanska et al., 2015; Dauphin et al., 2014). This limitation leads to reduced approximation accuracy (see Figure 1) and suboptimal performance in real-world applications (see Sec. 5). Additionally, the need to compute the inverse Hessian constrains the scalability of these methods for large-scale applications.

**TracIn.** Recently, another notable work, TracIn (Pruthi et al., 2020; Xia et al., 2024), bypassed the convex loss assumption by introducing a heuristic proxy metric to the original influence metric defined in Eq.1, called $\texttt{TracInIdeal}(z, \mathbf{V}) = \sum_{t:z_t=z} \mathcal{L}(\mathbf{V}, \theta^t) - \mathcal{L}(\mathbf{V}, \theta^{t+1})$. This metric measures the total reduction in loss on the validation set $\mathbf{V}$ caused by the stochastic gradient descent process whenever the training example $z$ is used. TracIn approximates this heuristic proxy with an efficient first-order estimator:

$$\text{TracIn}(z, \mathbf{V}) = \sum_{t \in \mathcal{T}_m} \eta_t \langle \nabla\mathcal{L}(\mathbf{V}, \theta^t), \nabla\mathcal{L}(\mathbf{D}, \theta^t) \rangle, \tag{3}$$

where $\langle \cdot, \cdot \rangle$ denotes the inner-product operation, $\eta_t$ is the learning rate at the $t$-th iteration, and $\mathcal{T}_m = t_1, ..., t_m$ is a set of sampled time steps.

Due to its first-order approximation, TracIn is more scalable for large-scale datasets than the method proposed by Koh and Liang (Koh & Liang, 2017). However, its heuristic goal differs from the original definition, and when combined with approximation errors, it often leads to imprecise estimates (Kwon et al., 2023). As shown in Fig. 1, the approximation accuracy of TracIn decreases as the dataset size and the number of model parameters increase. Furthermore, this approach is not suitable for estimating a sample $z$'s influence on the parameters.

# 3 INFLUENCE ESTIMATION VIA DIFFERENTIAL APPROXIMATION

In the following sections, we first explain how to expand the influence as the cumulative sum of differences in Sec. 3.1 and introduce an efficient second-order estimator for the difference term (**Lemma** 3.1) along with its first-order approximations (Eq. 6). Next, building on the derived difference term, we present Diff-In, which estimates the influence on parameters and loss values in Proposition 3.2. Finally, we discuss implementation details in Sec. 3.2. We will extend the method to other optimizers in Sec. B.

## 3.1 DIFF-IN

Here, we formulate the influence as the cumulative sum of its differences between successive training time steps. We denote the influence of sample $z$ on the parameters at the $t$-th training iteration as $\mathcal{I}_\theta^t(z) = \theta_{-z}^t - \theta^t$. The sample-wise influence difference between adjacent training steps is given by $\mathcal{D}^t(z) = \mathcal{I}_\theta^{t+1}(z) - \mathcal{I}_\theta^t(z)$. Given that the maximum number of iterations is $T$, we have $\mathcal{I}_\theta = \mathcal{I}_\theta^T$. Therefore, we can express $\mathcal{I}_\theta^t$ as follows:

$$\mathcal{I}_\theta(z) = \sum_{t<T} \left( \mathcal{I}_\theta^{t+1}(z) - \mathcal{I}_\theta^t(z) \right) + \mathcal{I}_\theta^0(z) = \sum_{t<T} \mathcal{D}^t(z) + \mathcal{I}_\theta^0(z) = \sum_{t<T} \mathcal{D}^t(z), \tag{4}$$

where $\mathcal{I}_\theta^0(z) = \theta_{-z}^0 - \theta^0 = 0$, since the removal or inclusion of sample $z$ does not affect the initial model parameters. Although simple, this approach allows us to approximate the difference terms without this approach enables us to approximate the difference terms without relying on convexity assumptions, as demonstrated below, thereby enhancing approximation accuracy.

**Estimation for the difference term** $\mathcal{D}^t(z)$  We can express $\mathcal{D}^t(z)$ as: $\mathcal{D}^t(z) = (\theta_{-z}^{t+1} - \theta_{-z}^t) - (\theta^{t+1} - \theta^t)$. Assuming the use of the SGD optimizer, according to the parameter update rule, we have $\theta^{t+1} - \theta^t = -\eta_t G^t$, where $G^t$ represent the gradient at iteration $t$ and $\eta_t$ denotes the learning rate. We can then approximate $\mathcal{D}^t(z) = -\eta_t(G_{-z}^t - G^t)$ where $G_{-z}^t$ denotes the gradient at iteration $t$ with sample $z$ removed from the training set. It is important to highlight that our approximation can be readily extended to other optimization methods by adjusting the parameter update rule as necessary (see Sec. B). By furthering expressing $G_{-z}^t - G^t$ using a continuous time approximation and introducing a perturbation parameter $\epsilon$ for the gradient difference terms, following (Koh & Liang, 2017), we can derive $\mathcal{D}^t(z)$ using second-order and first-order terms through Taylor's expansion. This leads us to the following Lemma.

**Lemma 3.1.** *Given the parameters $\theta_D^t$ optimized via the SGD optimizer at the $t$-th iteration, by supposing the time-step to be continuous, $\mathcal{D}^t(z)$ can be approximated as follow,*

$$\hat{\mathcal{D}}^t(z) = \sum_{k \leqslant t} a_{t,k} \left( H_{B^k}^k G_z^k + H_z^k G_{B^k}^k \right), \tag{5}$$

*where $B^t \subset \mathbf{D}$ is the training mini-batch at the $t$-th, $H_{B^t}^t = \nabla^2 \mathcal{L}(B^t, \theta^t)$ is the hessian over the batch, $G_{B^t}^t = \nabla \mathcal{L}(B^t, \theta^t)$ is the gradient over the batch. $H_z^t = \nabla^2 \ell(z, \theta^t)$ and $G_z^t = \nabla \ell(z, \theta^t)$ are the hessian and the gradient on the sample $z$, respectively. The coefficient $a_{t,k} = -(\eta_t \eta_k)^2 / N$ is a function of the learning rate $\eta_t$ and $\eta_k$.*

The detailed proof is provided in Appendix F.1. Despite being a second-order estimator, the Hessian-gradient product can be efficiently approximated using the classic finite difference method (Pearlmutter, 1994), as described by the following rule:

$$HG \approx \lim_{\epsilon \to 0} [\nabla \mathcal{L}(\theta + \epsilon G) - \nabla \mathcal{L}(\theta)] / \epsilon, \tag{6}$$

where the complexity is $\mathcal{O}(p)$, with $p$ representing the number of parameters, making it comparable to first-order methods. This approximation can be easily implemented and efficiently executed using existing deep learning frameworks like PyTorch (Adam Paszke et al., 2017).

**Diff-In: influence on parameters and influence on loss.**  Using the approximation for the difference term of influence on parameters, $\hat{\mathcal{D}}^t(z)$, as shown in Eq. 5, we can calculate the influence on parameters by accumulating this term, as described in Eq. 4. Similarly, for the influence on the

loss $\mathcal{I}(z, \mathbf{V})$, we expand it by accumulating its differences over different time steps. By applying first-order approximations of the loss and accounting for the influence of a sample $z$ on model parameters, $D^t(z)$, at different time steps $t$, we find $\mathcal{I}(z, \mathbf{V})$ can be computed by accumulating the product of gradient of validation loss, $\nabla \mathcal{L}(\mathbf{V}, \theta^t)$, and the difference term of influence on parameters, $\hat{\mathcal{D}}^t(z)$:

**Proposition 3.2.** *Let $\langle \cdot, \cdot \rangle$ denote the inner-product operation. By using the estimation for $\mathcal{D}^t(z)$ in Lemma 3.1 and keeping the symbol convention aforementioned, the differential influence function calculates the influence on parameters $\mathcal{I}_\theta(z)$ and the influence on loss $\mathcal{I}(z, \mathbf{V})$ as:*

$$\mathcal{I}_\theta(z) = \sum_t \hat{\mathcal{D}}^t(z), \quad \mathcal{I}(z, \mathbf{V}) = \sum_t \left\langle \nabla \mathcal{L}(\mathbf{V}, \theta^t), \ \hat{\mathcal{D}}^t(z) \right\rangle. \tag{7}$$

The proof of the above proposition is provided in Appendix F. It is important to note that the naive calculation of this approximation is inefficient. This is because both the difference term in Eq. 5 and the final Diff-In estimator in Eq. 7 require summing over historical time steps, which is impractical for real-world applications. To address this issue, in the next subsection, we introduce practical techniques involving the use of checkpoints to significantly reduce computational costs.

## 3.2 Practical Implementation Using Checkpoints

Here, we present practical methods commonly used in practice to accelerate influence estimation. First, for the estimator for the influence difference term in Eq. 5, we calculate it using the $t$-th checkpoint rather than all $k \leqslant t$ learning steps:

$$\hat{\mathcal{D}}^t(z) \approx \frac{-t(\eta^t)^2}{N} \left( H_{B^t}^t G_z^t + H_z^t G_{B^t}^t \right). \tag{8}$$

This is equivalent to when we use sampling to estimate the summation over time steps and only take the very last time step ($k = t$) as the sampled time point. We have discovered that this extremely simple operation is rather effective in the experiments. The practical performance of this strategy is better and more stable than sampling five random time steps. See Figure 2!

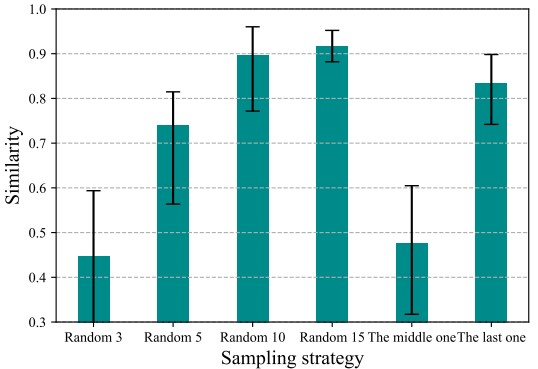

Figure 2: Comparison of the (Cosine) similarity between the estimated difference in Eq 5 via different sampling strategies and the ground truth obtained by retraining. *[Random n]* means randomly selecting $n$ time steps. *[The middle one]* indicates set $k = t//2$. *[The last one]* indicates set $k = t$ as Eq. 8. This experiment is done on CIFAR-10 with ResNet-18 where $t = 37$.

Second, computing Eq. 7 requires revisiting all learning steps, which is impractical in practice. To address this, we adopt two practical strategies. Following the approach in (Pruthi et al., 2020; Tan et al., 2023), we calculate the influence using saved intermediate checkpoints rather than all learning steps. By applying the efficient empirical rule in Eq.8, we have the influence calculation with checkpoints:

$$\mathcal{I}_\theta(z) = \frac{1}{m} \sum_{t \in \mathcal{T}_m} \frac{-t(\eta^t)^2}{N} \left( H_{B^t}^t G_z^t + H_z^t G_{B^t}^t \right),$$

$$\mathcal{I}(z, \mathbf{V}) = \frac{1}{m} \sum_{t \in \mathcal{T}_m} \frac{-t(\eta^t)^2}{N} \left\langle \nabla \mathcal{L}(\mathbf{V}, \theta^t), \ \left( H_{B^t}^t G_z^t + H_z^t G_{B^t}^t \right) \right\rangle, \tag{9}$$

where $\mathcal{T}_m = \{t_1, ..., t_m\}$ is a set of randomly selected time-steps. Since the calculation requires access to the batch of data $B^t$ used during training, which is often unavailable, we instead sample a random batch of data for the calculation, following (Pruthi et al., 2020). The pseudocode for Diff-In is provided in Appendix A.

The impact of the hyper-parameter $m$ is explored in Sec. 5.4. As demonstrated in the experiments in Sec.5.4, we found that a relatively small number of sampled time steps is sufficient to achieve

strong performance. A similar sampling strategy could be seen in (Pruthi et al., 2020; Tan et al., 2023; Ghorbani & Zou, 2019).

The naive calculation for Diff-In in Proposition 3.2 has a complexity of $\mathcal{O}(T^2 p)$, where $T$ is the total number of training and $p$ is the number of parameters. However, since $T$ can be very large, tracing every time step would be computationally expensive and resource-intensive. To address this, we introduce the aforementioned time-step sampling strategy, and the overall complexity is reduced to $\mathcal{O}(mp)$.

## 4 APPROXIMATION ERROR

We analyze the approximation error of the proposed influence estimator in Proposition 3.2, where $\mathcal{I}(z)$ is the estimated influence and $\mathcal{I}^{\text{Exact}}(z)$ is the exact influence calculated by the vanilla leave-one-out retraining:

**Proposition 4.1.** *Supposing the loss has $\ell$-Lipschitz continuous gradient and the gradient norm of the parameter is upper-bounded by $g$, and assuming the learning rate $\eta \leqslant 1$ and the momentum weight $\beta \leqslant 1$ (if available), the error between the approximated $\mathcal{I}(z)$ and the exact $\mathcal{I}^{\text{Exact}}(z)$ is bounded by:*

$$|\mathcal{I}(z) - \mathcal{I}^{\text{Exact}}(z)| \leqslant 2T^2 \ell(T+1)C + \frac{T^2}{N}g, \qquad (10)$$

*where $|\cdot|$ is the norm, and $T$ is the maximum iteration, $N$ is the number of training samples, $C$ represents the farthest distance the neural network parameters move away from their initial state during training when any subset $\mathbf{D}_s \subset \mathbf{D}$ is used as the training set, under the fixed environmental settings, initialization and training strategy, that is, $C = \max_{\mathbf{D}_s \subset \mathbf{D}, t \leqslant T} |\theta^t_{\mathbf{D}_s} - \theta^0|$. Note that $C$ has a polynomial growth with $T$, e.g. when using the SGD optimizer, $C \leqslant Tg$, where $C$ is less than the product of the number of time steps $T$ and the upper bound of the gradient norm $g$.*

The detailed derivation is provided in Appendix G. This Proposition highlights several key points: (1) Approximation becomes more challenging as the number of iterations increases, since the approximation error grows polynomially with the number of time steps. This aligns with findings in previous works (Schioppa et al., 2024; Tan et al., 2023). (2) The approximation accuracy of Diff-In is superior to that of earlier approaches such as (Hara et al., 2019; Schioppa et al., 2024), where the error exhibited exponential growth with increasing training iterations. (3) Models with smoother loss surfaces (corresponding to smaller $g$ and $\ell$) will be easier to obtain better approximate accuracy.

## 5 EXPERIMENTS

We extensively evaluate and benchmark our method on three major data-centric tasks: dataset cleaning (Sec. 5.1), data deletion (Sec. 5.2), and coreset selection (Sec. 5.3). Then, we conduct ablation studies and analysis on Diff-In (Sec. 5.4). All the results are averaged over 5 independent runs. More details on the implementation and experiments are reported in the Appendix D.

### 5.1 DATA CLEANING: FINDING THE WRONGLY LABELED SAMPLES

**Setup.** Data cleaning (Csaba Kertész, 2021; Tang et al., 2021) is a crucial step in the data pre-processing pipeline aimed at improving the quality and reliability of dataset labels. In our experiments, we define data cleaning as the process of identifying mislabeled samples within the dataset. Detailed experimental settings can be found in Appendix E.

We assess label quality by examining a sample's influence on its own loss (Pruthi et al., 2020). According to the definition of a single data point's influence on the loss function (Eq. 1), setting the validation set as the sample $z$ itself, i.e., $V = z$, yields an important self-influence metric, $\mathcal{I}(z, z)$:

$$\mathcal{I}(z, z) = \mathcal{L}(z, \theta^*_{-z}) - \mathcal{L}(z, \theta^*), \qquad (11)$$

which represents the influence of a training point $z$ on its own loss. Self-influence plays a key role in identifying mislabeled points (Pruthi et al., 2020; Koh & Liang, 2017). Outliers or mislabeled data can significantly impact their own loss, or self-influence (Pruthi et al., 2020), because the model may

Table 1: Experimental results of data cleaning, aiming to identify the noisy data of the given dataset. The performance metric is the Precision in Eq. 12. The best results are bolded. Please check Appendix D.3 for std information. In Table.12, we present more experimental results on GSM8K for IP (Yang et al., 2024) and OGI (Chhabra et al., 2024a).

| Dataset ($\rightarrow$) | SVHN | | | | | | Tiny-ImageNet | | | | | | GSM8K | | | | | |
|---|---|---|---|---|---|---|---|---|---|---|---|---|---|---|---|---|---|---|
| Selection Rate ($\rightarrow$) | 20% | 30% | 40% | 50% | 60% | 70% | 20% | 30% | 40% | 50% | 60% | 70% | 20% | 30% | 40% | 50% | 60% | 70% |
| Random | 20.0 | 30.0 | 40.0 | 50.0 | 60.0 | 70.0 | 20.0 | 30.0 | 40.0 | 50.0 | 60.0 | 70.0 | 20.0 | 30.0 | 40.0 | 50.0 | 60.0 | 70.0 |
| Loss value | 27.2 | 43.4 | 51.0 | 65.9 | 69.1 | 72.2 | 21.2 | 31.7 | 42.1 | 52.2 | 62.6 | 65.3 | 28.4 | 39.3 | 51.5 | 57.4 | 63.7 | 69.4 |
| IF | 41.6 | 46.3 | 59.5 | 66.8 | 72.5 | 78.1 | 21.1 | 31.6 | 42.0 | 52.2 | 62.3 | 69.4 | 67.2 | 71.4 | 79.6 | 88.5 | 94.7 | 97.7 |
| DataInf | 42.9 | 46.6 | 61.9 | 68.5 | 72.9 | 77.6 | 26.9 | 38.6 | 45.5 | 58.6 | 64.8 | 70.5 | 68.5 | 73.4 | 84.2 | 90.1 | 96.2 | 98.5 |
| TracIN | 81.4 | 90.2 | 95.5 | 98.2 | 99.2 | 99.6 | 76.3 | 91.0 | 95.9 | 97.8 | 98.4 | 99.3 | 78.4 | 84.5 | 92.2 | 99.2 | 99.2 | 100.0 |
| GEX | 80.7 | 84.9 | 86.7 | 89.5 | 91.5 | 94.5 | 72.7 | 77.2 | 81.9 | 87.3 | 90.1 | 92.2 | 77.2 | 87.1 | 91.6 | 95.3 | 99.9 | 100.0 |
| Diff-In | **90.2** | **96.4** | **98.7** | **99.6** | **99.6** | **99.9** | **88.2** | **98.0** | **98.6** | **99.0** | **99.8** | **100.0** | **86.1** | **92.2** | **99.3** | **99.9** | **99.9** | **100.0** |

not receive sufficient information from other samples to handle these outliers effectively, leading to higher self-influence values for such samples.

**Experiments.** We perform experiments on two image classification datasets, namely SVHN (Netzer et al., 2011) and Tiny-ImageNet (Ya Le & Xuan S. Yang, 2015). Additionally, we also conduct experiments with the GSM8K (Cobbe et al., 2021) dataset for large language models (Llama Team, 2024). To create a noisy dataset, we randomly select 20% of the data and perturb their corresponding labels by following (Pruthi et al., 2020; Kim et al., 2024). The experimental results are presented in Table 1, where we report the Precision metric under different selection ratio settings,

$$\texttt{Precision} = N_{\texttt{found}}/N_{\texttt{all}}, \tag{12}$$

where $N_{\texttt{found}}$ indicates the number of noise samples found by algorithms and $N_{\texttt{all}}$ is the number of all noise samples. Detailed introduction for settings and datasets can be seen in Appendix E.1 and E.3. For experiments on SVHN (Netzer et al., 2011) and Tiny-ImageNet (Ya Le & Xuan S. Yang, 2015), we train ResNet-18 models from scratch. For GSM8K, we LoRA-finetune a Llama 3.1 8B model (Llama Team, 2024) on GSM8K (Cobbe et al., 2021) and then use the parameters of LoRA to calculate influence by following (Xia et al., 2024; Kwon et al., 2023). Other influence estimators, such as IF (Koh & Liang, 2017), TracIn (Pruthi et al., 2020), GEX (Kim et al., 2024), and DataInf (Kwon et al., 2023), are also employed to compute the self-influence metric. We additionally take the loss value for data cleaning as a classical metric. The greater the loss, the more probable it is that a sample is a noise sample.

Diff-In outperforms the compared methods in all scenarios. For instance, considering the challenging Tiny-ImageNet dataset, where both the IF and entropy-based methods only achieve marginal improvements over random selection. When employing Diff-In to select the top 20% samples, it is possible to identify over 88% of the erroneous samples, exceeding others like TracIN by approximately 12.0%. For GSM8K, when the selection rate is 20%, using Diff-In surpasses the second-best method, TracIN, by around 9.0%. Refer to Appendix D.2 for more experiments.

## 5.2 DATA DELETION: REMOVING THE INFLUENCE OF NOISY DATA WITHOUT RETRAINING

**Setup.** Data deletion, also known as machine unlearning (Fu et al., 2022), is the task of eliminating the impact of certain training data items from a learned model. It is noteworthy that some methods in this field rely on retraining, while others do not. Diff-In falls into the latter category. To ensure fairness, the baselines discussed here exclusively consist of methods that do not depend on retraining. Herein, we employ influence estimators to eliminate the impact on the learned model by noisy samples. We introduce 20% label noise into the dataset and train a ResNet-18 model. Note that this experiment is different from the Data Cleaning shown above. Data Cleaning aims to find mislabeled data, while the purpose

Table 2: The experimental results of data deletion, where the column "Noise" contains results on the noise set while the column "Oracle" contains results retraining on the cleaned set (all noise samples are removed). The performance metric is Accuracy@1.

| Dataset ($\rightarrow$) | SVHN | CIFAR-100 | Tiny-ImageNet |
|---|---|---|---|
| Noise | 77.4 | 64.5 | 39.7 |
| Oracle | **83.6** | **74.1** | **45.7** |
| IF (Koh & Liang, 2017) | $79.2_{+1.86}$ | $65.1_{+1.92}$ | $38.2_{+2.25}$ |
| MC-IF (Fu et al., 2022) | $80.7_{+0.94}$ | $71.9_{+1.26}$ | $40.3_{+1.23}$ |
| SEEE (Peste et al., 2021) | $81.4_{+0.82}$ | $72.2_{+1.71}$ | $41.5_{+1.04}$ |
| Diff-In | $83.0_{+0.71}$ | $73.5_{+0.85}$ | $42.9_{+1.04}$ |

of Data deletion here is: given that the model has already been trained on a dataset, use some methods to eliminate the influence of particular data samples without retraining. Detailed experimental settings can be seen in Appendix E.

**Deletion with influence functions.** Considering the physical meaning of the influence $\mathcal{I}_\theta(z) = \theta^*_{-z} - \theta^*$, we can approximately obtain the leaving-one-retraining parameters by $\theta^*_{-z} = \mathcal{I}_\theta(z) + \theta^*$ after calculating the approximation $\mathcal{I}_\theta(z)$. Hence, $\theta^*_{-z}$ is the approximated parameter after performing data deletion. For deleting a set of samples $\mathbf{Z}$ from the model, according to the additivity assumption of influence functions Koh & Liang (2017); Koh et al. (2019); Yang et al. (2023); Tan et al. (2023), we can add up their influence as a whole, specifically, $\theta^*_{-\mathbf{Z}} = \sum_{z \in \mathbf{Z}} \mathcal{I}_\theta(z) + \theta^*$. Note that many methods (Pruthi et al., 2020; Kim et al., 2024; Ko et al., 2024; Kwon et al., 2023) do not provide the influence on parameters, $\mathcal{I}(\theta)$, and therefore cannot be applied in this experiment (see Table 6). To execute data deletion for noise samples, we compute the influence of those noisy data on the parameter denoted by $\mathcal{I}_\theta(\mathbf{D}_n)$, where $\mathbf{D}_n$ signifies the subset encompassing all the noisy data.

**Experiments.** For comparison, apart from the simplistic retraining method (referred to as the Oracle), we also engage in a comparison with the Influence Function estimator (abbreviated as IF) put forward by Koh and Liang (Koh & Liang, 2017) and the MCMC-IF (Fu et al., 2022), SEEE (Peste et al., 2021) as baseline approaches. The experiments are carried out on SVHN (Netzer et al., 2011), CIFAR-100 (Alex Krizhevsky, 2009), and Tiny-ImageNet (Ya Le & Xuan S. Yang, 2015). The experimental results are presented in Table 2. Diff-In attains a superior performance of 2.0% compared to MCMC-IF (Fu et al., 2022). It is noteworthy that the performance disparities to the Oracle brute-force retraining on SVHN and CIFAR-100 are approximately 0.6%.

## 5.3 CORESET SELECTION

**Setup.** Coreset selection aims to identify a compact subset of the training data in such a way that the model's performance on this subset closely approximates its performance on the entire dataset (Ozan Sener & Silvio Savarese, 2018; Sorscher et al., 2022). The proportion of the selected subset to the original dataset is termed the selection ratio. We carried out experiments on both an image classification task and a vision-language pretraining task. The detailed experimental setups can be found in Appendix E. For all experiments, we first pre-train a model on the full training set, then calculate the influence score for each training sample on the training loss (Koh & Liang, 2017; Kim et al., 2024; Kwon et al., 2023), that is, $\mathcal{I}(z, \mathbf{D})$. A higher influence score on the training loss indicates that a sample may have a more positive influence (Tan et al., 2023). Finally, we retain the subset corresponding to the highest influence scores as the coreset. Further particulars regarding the experimental setups are provided in Appendix E. In Table.11, we present more experimental results on the coreset for supervised-finetuning Llama-3-8B on OpenMathInstruct (Toshniwal et al., 2024).

**Image classification.** We carried out experiments on three public benchmarks: CIFAR-100 (Alex Krizhevsky, 2009), Tiny-ImageNet (Ya Le & Xuan S. Yang, 2015), and ImageNet-1K (Russakovsky et al., 2015). In these experiments, we compared the performance of Diff-In with various influence estimators, including IF (Koh & Liang, 2017), TracIn (Pruthi et al., 2020), GEX (Kim et al., 2024), and DataInf (Kwon et al., 2023). Notably, to mimic real-world limitations on computational resources, the pre-trained model was trained for only a few epochs. Additionally, we compared Diff-In to state-of-the-art coreset selection methods such as Prototypicality Coreset (also termed SSP) (Sorscher et al., 2022) and Moderate Coreset (Xia et al., 2023).

The experimental results are presented in Table 3. Among all the influence estimators, Diff-In consistently achieves the best results across most settings and datasets. This superiority is especially prominent at lower selection ratios. For instance, on Tiny-ImageNet, Diff-In outperforms the best baseline estimator (TracIn) by 2.4% at a selection rate of 20%. On ImageNet, Diff-In surpasses TracIn by 3.9% at a selection rate of 30%. Moreover, even when contrasted with methods specifically designed for coreset selection such as SSP and Moderate Coreset, Diff-In remains highly competitive, sustaining superior performance in nearly all settings—except for the 70% setting on CIFAR-100 and Tiny-ImageNet and the 40% setting on ImageNet-1K.

**Vision language pretraining.** We carried out experiments on the vision-language dataset CC12M (Soravit Changpinyo et al., 2021), which encompasses 12 million image-text pairs sourced from the Internet, for CLIP-like vision-language pre-training. The pre-trained CLIP model (Alec Radford et al., 2021) is frequently employed to rate image-text data from the vision-language dataset, wherein

Table 3: Experimental results of coreset selection on three image classification datasets with ResNet-50 model. The best results are bolded for baselines and ours, respectively. The performance metric is Accuracy@1. Please check Appendix D.3 for std information.

| Dataset (→) | CIFAR10 | | | | | | Tiny-ImageNet | | | | | | ImageNet-1K | | | | | |
|---|---|---|---|---|---|---|---|---|---|---|---|---|---|---|---|---|---|---|
| Selection Rate (→) | 20% | 30% | 50% | 70% | 80% | 100% | 20% | 30% | 50% | 70% | 80% | 100% | 20% | 30% | 50% | 70% | 80% | 100% |
| **Random** | 50.2 | 53.6 | 64.3 | 71.0 | 74.1 | 78.1 | 24.0 | 29.7 | 34.4 | 40.9 | 45.7 | 49.3 | 61.6 | 65.9 | 67.7 | 70.3 | 72.9 | 76.4 |
| **SSP** | 44.4 | 54.6 | 62.9 | 70.7 | 75.2 | - | 20.8 | 27.6 | 32.5 | 39.6 | 44.9 | - | 31.2 | 51.4 | 60.3 | 69.8 | **75.5** | - |
| **Moderate** | 51.8 | 57.7 | 64.9 | **71.8** | 74.2 | - | 25.2 | **30.5** | 34.8 | **41.4** | 46.0 | - | 61.1 | 67.8 | **70.0** | 73.0 | 75.3 | - |
| **IF** | 26.4 | 36.1 | 47.1 | 51.2 | 63.5 | - | 14.3 | 19.1 | 24.5 | 31.2 | 37.9 | - | 25.6 | 30.5 | 49.9 | 56.7 | 68.1 | - |
| **DataInf** | 26.9 | 35.8 | 48.2 | 52.5 | 65.5 | - | 16.1 | 21.2 | 27.7 | 33.1 | 42.4 | - | 27.7 | 32.3 | 52.6 | 58.2 | 67.0 | - |
| **TracIn** | 50.4 | 58.4 | 63.9 | 70.4 | 72.3 | - | 24.2 | 28.8 | 35.3 | 40.1 | 45.8 | - | 61.3 | 64.4 | 70.3 | 71.4 | 73.6 | - |
| **GEX** | 44.8 | 50.0 | 57.5 | 62.0 | 69.9 | - | 20.5 | 23.0 | 29.4 | 36.6 | 41.6 | - | 55.3 | 58.7 | 65.7 | 69.8 | 70.1 | - |
| **Diff-In** | **52.0** | **59.4** | **65.7** | 71.5 | **75.3** | - | **26.6** | **30.5** | **36.0** | 40.9 | **46.4** | - | **61.7** | **68.3** | 69.8 | **73.6** | **75.5** | - |

Table 4: Experimental results of coreset selection on vision-language pertaining. Regarding Diff-In, we contemplate two time-step configurations: randomly selecting times (designated as Diff-In, the normal setting) and selecting only the final time step (designated as Diff-In-F, the efficient setting). The top-2 highest results are bolded.

| Dataset (→) | Zero-shot Classification (Acc@1) | | | | | I2T Retrieval (Recall@1) | | | | | T2I Retrieval (Recall@1) | | | | | Linear Prob (Acc@1) | | | | |
|---|---|---|---|---|---|---|---|---|---|---|---|---|---|---|---|---|---|---|---|---|
| Selection Rate (→) | 10% | 20% | 40% | 60% | 100% | 10% | 20% | 40% | 60% | 100% | 10% | 20% | 40% | 60% | 100% | 10% | 20% | 40% | 60% | 100% |
| **Random** | 19.7 | 21.3 | 25.3 | 29.0 | 37.2 | 29.9 | 32.1 | 38.1 | 42.3 | 51.7 | 21.2 | 23.1 | 25.4 | 28.6 | 44.5 | 41.6 | 45.4 | 51.3 | 59.1 | 67.5 |
| **SSP** | 18.0 | 20.4 | 23.2 | 28.4 | - | 24.5 | 31.8 | 35.0 | 39.4 | - | 21.1 | 22.5 | 24.8 | 29.2 | - | 40.8 | 44.6 | 52.0 | 58.4 | - |
| **CLIP-score** | 24.1 | 27.4 | 32.4 | 34.1 | - | 33.9 | 39.1 | 45.2 | 47.1 | - | **23.7** | 28.4 | 36.4 | 40.3 | - | 47.4 | 52.9 | 59.3 | 62.1 | - |
| **Moderate** | 24.7 | 27.7 | 31.7 | 32.0 | - | 33.4 | 37.5 | 41.0 | 43.7 | - | 22.7 | 25.8 | 34.2 | 38.9 | - | 47.9 | 51.4 | 56.5 | 60.2 | - |
| **IF** | 22.3 | 24.8 | 29.0 | 30.4 | - | 30.4 | 37.3 | 42.5 | 45.9 | - | 20.8 | 25.2 | 30.4 | 35.7 | - | 43.3 | 47.1 | 54.7 | 60.7 | - |
| **DataInf** | 22.5 | 25.4 | 29.8 | 31.3 | - | 30.1 | 37.2 | 43.3 | 46.7 | - | 21.5 | 25.9 | 31.9 | 35.6 | - | 43.8 | 48.3 | 55.1 | 59.4 | - |
| **GEX** | 22.2 | 27.5 | 30.6 | 33.3 | - | 31.6 | 37.9 | 42.9 | 47.1 | - | 21.9 | 26.0 | 35.9 | 39.3 | - | 43.1 | 51.4 | 57.1 | 61.6 | - |
| **TracIn** | **24.7** | 27.1 | 31.9 | **34.5** | - | 33.1 | **39.5** | 45.5 | 47.8 | - | 22.4 | 28.1 | 36.7 | 40.8 | - | **47.8** | 53.0 | 58.9 | **63.0** | - |
| **Diff-In-F** | 24.3 | **27.8** | **32.9** | 34.4 | - | 34.2 | 39.5 | 45.4 | 47.7 | - | 23.5 | **29.0** | **37.1** | 40.7 | - | **47.8** | **53.1** | **59.5** | 62.5 | - |
| **Diff-In** | **25.8** | **28.3** | **33.5** | **36.2** | - | **34.4** | **40.6** | **46.4** | **48.9** | - | **23.9** | **29.7** | **38.0** | **41.1** | - | **48.3** | **53.7** | **60.2** | **63.4** | - |

a higher CLIP score denotes superior image-text alignment. Consequently, there exists a prevalent and valuable coreset baseline (Christoph Schuhmann et al., 2022; Li et al., 2022; 2023) for vision-language datasets that retain data with higher CLIP scores. In addition to those influence-based approaches employed in the image classification task, we consider another baseline, the **Moderate** coreset (Xia et al., 2023) over CLIP scores: Selecting those samples with the median score since the highly-scored samples might be the too-easy samples.

For all methods, we train a CLIP-Vit-Base model on CC12M; the detailed settings are introduced in the appendix. Regarding Diff-In, we use two time-steps: randomly selecting times (designated as Diff-In) and selecting only the final time step (designated as Diff-In-F). After data selection, we conduct CLIP pre-training on the coreset and evaluate the model on four downstream tasks: Zero-shot ImageNet Classification, Image-to-Text Retrieval, and Text-to-Image Retrieval on Flickr30K (Plummer et al., 2015), and Linear Probing on ImageNet; refer to Appendix E.2 for more particulars.

The experimental results are shown in Table 4. For all 12M samples, it takes about 9.4 minutes to calculate CLIP scores, 10.2 minutes to calculate Diff-In-F, and 59.2 minutes to calculate Diff-In. Diff-In consistently outperforms all baselines across all selection ratios, surpassing the CLIP score by 1%-2% in most cases. Notably, with a similar time cost as the popular CLIP score, Diff-In-F performs better than the CLIP score on all experiments except the (selection-rate=10%) T2I retrieval experiment. This is because CLIP scores focus solely on evaluating text-image alignment and favor simpler, more obvious matches, potentially overlooking more complex but informative samples. By considering influence, Diff-In offers a more comprehensive evaluation, considering a sample's alignment and informativeness.

## 5.4 ABLATION STUDY

Here we conduct ablation studies for the effect of the number of sampled time steps on the final performance and speed, and the generalization ability to various optimizer choices. In Table.11, we present more ablation results for the choice of the number of sampled time steps and the speed test on the coreset for supervised-finetuning Llama-3-8B on OpenMathInstruct (Toshniwal et al., 2024).

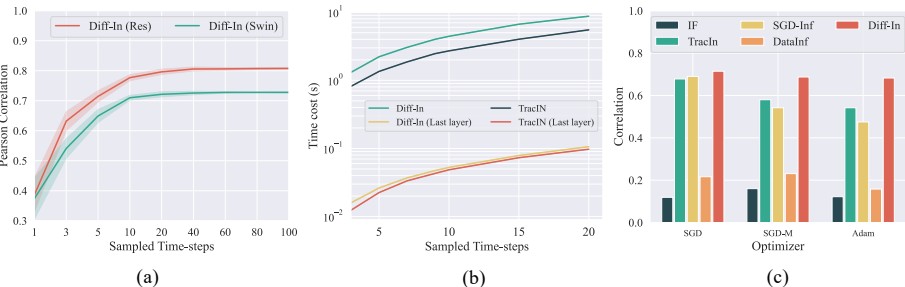

(a)  (b)  (c)

Figure 3: In (a), on CIFAR-10, we study the effect of the number of sampled time steps, where "Res" refers to the ResNet-18 (He et al., 2016) and "Swin" refers to the Swin-Transformer (Tiny) (Liu et al., 2021a). In (b), on CIFAR-10 and with ResNet-18, the results show the practical time consumption comparison, where "(Last layer)" refers to the computation that only takes into account the parameters of the last layer, otherwise, all the parameters are considered. In (c), on CIFAR-10 and with ResNet-18, the figure compares the approximation precision of our Diff-In and other baselines under different optimizer settings, namely the vanilla SGD optimizer, the SGD optimizer with a momentum weight of 0.9, and the Adam optimizer.

**Study of sampled time steps.** For the Diff-In with checkpoints in Sec. 3.2, the larger $m$ (the number of sampled time steps), the closer the estimation in Proposition 3.2, but also the higher the computation cost. Here, we study this by measuring the Pearson correlation between the approximation and the exact brute-force retraining influence under different settings for $m \in \{2, 3, 4, 5, 10, 20, 40, 60, 100\}$. The experiment is conducted on the CIFAR-10 dataset (Alex Krizhevsky, 2009) using ResNet-18 (He et al., 2016) and Swin-Transformer (Liu et al., 2021a) with the vanilla SGD optimizer. Fig. 3 (a) visualize the results with std bar. In conclusion, the performance growth will gradually slow down when $m$ exceeds 5 and eventually converges. Hence, we recommend $m = 5$ for most settings.

**Speed comparison.** Figure 3(b) compares the practical computation speed of our Diff-In with TracIn (Pruthi et al., 2020). Here, we consider settings for the calculation: using all parameters (Koh & Liang, 2017), or only using the parameters in the last layer (Yang et al., 2023; Tan et al., 2023). The experiment is conducted on the CIFAR-10 dataset (Alex Krizhevsky, 2009) using ResNet-18 (He et al., 2016). With only the last layer's parameters and $m = 5$, Diff-In and TracIn can significantly speed up the calculation by two orders of magnitude and achieve similar speeds. Moreover, by setting $m = 5$, the time cost can be further reduced to 0.02 sec for Diff-In. Note that the classic influence estimator proposed by Koh and Liang (Koh & Liang, 2017) costs around 3 sec and 0.7 sec with all parameters and the last layer's parameters respectively, which is not displayed in the figure.

**Approximation precision comparison.** Here, we compare the precision of our approach with Koh's method (Koh & Liang, 2017) (abbreviated as IF), DataInf (Kwon et al., 2023), SGD-Inf (Hara et al., 2019), and TracIn (Pruthi et al., 2020), in terms of approximation precision. This study is conducted under three optimizer settings: vanilla SGD, SGD with momentum, and Adam (Kingma & Ba, 2015). Diff-In consistently maintains the highest level of approximation precision across all settings. Diff-In shows the leading performance, especially under momentum-based optimizer settings, where TracIn's and SGD-Inf's precision significantly decreases.

## 6 CONCLUSION

In conclusion, this paper has presented a novel formulation, Diff-In, which approximates a sample's influence by accumulating the differences in influence between consecutive learning steps. Through the use of second-order approximations, Diff-In has been able to achieve high accuracy in approximating the difference terms without the need for model convexity required by traditional methods. Despite its second-order nature, it maintains a computational complexity comparable to first-order methods, making it scalable and efficient, achieved by computing the product of the Hessian and gradient using finite differences of first-order gradients. Theoretically and empirically, Diff-In has shown remarkable performance. The theoretical analysis indicates its significantly lower approximation error compared to existing estimators, and extensive experiments across multiple benchmark datasets in various data-centric tasks such as data cleaning, data deletion, and coreset selection have further validated its superiority.

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

## LIMITATIONS

Like many existing methods, Diff-In focuses on sample-level influence and is currently limited in its ability to address more general applications, such as the influence of neural network parameters or hyperparameters. In future work, we plan to develop more comprehensive and unified theoretical tools to better understand the influence of various factors affecting neural network training.

## BROADER IMPACT

This paper presents a novel and effective influence analysis algorithm to advance the deep learning area. There are some potential positive societal effects, such as helping people better understand the role of data to develop more robust deep learning systems and possibly even be used to eliminate data bias. As for the potential negative impacts, we believe that this technology, and even the entire field of artificial intelligence, may be applied to inhumane social surveillance, which should be taken seriously by legislative bodies worldwide.

Table 5: Comprehensive overview of the notational convention.

| Notation | Description |
| --- | --- |
| $\mathbf{D}$ | The training set, and the size of the training set is $|\mathbf{D}| = N$. |
| $\langle \cdot, \cdot \rangle$ | The inner-product operator. |
| $z \in \mathbf{D}$ | A training data. |
| $\mathbf{D}/z$ | The training set excluded the sample $z$. |
| $\mathbf{V}$ | The validation set. |
| $\mathbf{B}_t$ | The mini-batch at the $t$-th iteration. |
| $\ell(\cdot)$ | The loss function over one data point. |
| $\mathcal{L}(\cdot)$ | The averaged loss over batch or set. |
| $\theta^*$ | The learned parameters optimized on the full training set after training. |
| $\theta^t$ | The learned parameters optimized on the full training set at the $t$-th iteration. |
| $\theta^*_{-z}$ | The learned parameters optimized on the dataset excluded the sample $z$ after training. |
| $\theta^t_{-z}$ | The learned parameters optimized on the dataset excluded the sample $z$ at the $t$-th iteration. |
| $\theta^*_{-\mathbf{Z}}$ | The learned parameters optimized on the dataset excluded a sample set $\mathbf{Z}$ after training. |
| $p$ | The number of trainable parameters. |
| $G$ | The gradient of the model parameter. |
| $H$ | The hessian of the model parameter. |
| $G^t_z$ | The gradient of the parameters at the $t$-th training iteration over the sample $z$. |
| $H^t_z$ | The Hessian of the parameters at the $t$-th training iteration over the sample $z$. |
| $G^t_{-z}$ | The gradient of the parameters at the $t$-th training iteration over $\mathbf{D}/z$. |
| $T$ | The maximum iteration of the training process. |
| $\mathcal{T}_m$ | A set of $m$ randomly selected time-steps $\mathcal{T}_m = \{t_1, ..., t_m\}$. |
| $m$ | The number of randomly selected time-steps in $\mathcal{T}_m$. |
| $\mathcal{I}_\theta(z),$ | The influences on parameters, also the Cook's distance (Cook, 1986). |
| $\mathcal{I}(z, \mathbf{V})$ | The influences on loss over the validation set $\mathbf{V}$. |
| $\mathcal{D}^t$ | The influence difference between two adjacent time steps $t$ and $t-1$. |
| $\eta$ | The learning rate in the optimizer. |
| $\beta$ | The momentum weight in the optimizer. |
| $\ell$ | The Lipschitz constant. |
| $g$ | The upper bound of the gradient norm. |
| $C$ | The farthest distance the neural network parameters move away from their initial state during training when any subset $\mathbf{D}_s \subset \mathbf{D}$ is used as the training set. |
| LOO | Leave-one-out. |

Table 6: Comparison of different methods. We compare several kinds of influence estimators, including, IF (Koh & Liang, 2017), DataInf (Kwon et al., 2023), GEX (Kim et al., 2024), and TracIn (Pruthi et al., 2020). Both IF (Koh & Liang, 2017) and DataInf (Kwon et al., 2023) require the loss convexity assumption and the stationarity assumption (the solution should be a stationary point). In this table, only TracIn and our Diff-In can be aware of training dynamics. Only our Diff-In has all the advantages, such as not requiring assumptions of convexity and stationarity, being able to consider the training dynamics, and being applicable for estimating the influence on parameters.

|  | IF | DataInf | GEX | TracIn | Diff-In |
|---|---|---|---|---|---|
| Avoids loss convexity | No | No | Yes | Yes | Yes |
| Avoids stationarity | No | No | Yes | Yes | Yes |
| Training dynamics | No | No | No | Yes | Yes |
| Influence on loss | Yes | Yes | Yes | Yes | Yes |
| Influence on parameters | Yes | No | No | No | Yes |

## A  PSEUDO-CODE OF DIFF-IN

We summarize the calculation in Algorithm 1. Given a selected training sample $z$, a validation set $\mathbf{V}$, and several time steps, we first randomly sample $m$ time steps from the training process and select the corresponding model checkpoints $(\theta_{\mathbf{D}}^t)_{t \in \{t_1, \ldots, t_m\}}$. Note that we can randomly select $m$ time steps before training and save the corresponding model checkpoints during training, without saving checkpoints at all steps. For each time-step $t$, we calculate the influence-difference term $\mathcal{D}^t(z)$ using Eq.4. Finally, after iterating over all the sampled time steps, it outputs the influence of the selected training sample $z$ on the model parameters and the loss on $\mathbf{V}$.

## B  HOW TO EXTEND DIFF-IN TO OTHER OPTIMIZERS?

Thus far, the method primarily addresses models optimized with standard SGD. However, thanks to our new formulation, it can be easily extended to momentum-based gradient descents, such as SGD-M, by adjusting the parameter update equation in the derivation. For instance, consider the general form of gradient descent with momentum: $\theta_{t+1} = \theta_t - \eta_t \left( (1 - \beta) G^t + \beta M_{t-1} \right)$. The corresponding generalized form of the estimator for $\hat{\mathcal{D}}^t(z)$ is:

$$\hat{\mathcal{D}}^t(z) = \sum_{k<t} \alpha_k^t \Big[ \sum_{q<k} H^q \sum_{e<q} \alpha_e^q \nabla \ell(z, \theta^e) + \sum_{q<k} \nabla^2 \ell(z, \theta^q) \sum_{e<q} \alpha_e^q G^e \Big],$$

where the coefficient $\alpha_k^t = \frac{1}{N} \left( \prod_{k<a<t} \eta_a \beta_1 \right) \eta_k (1 - \beta_1)$ is defined by the learning rate $\eta$ and the momentum weight $\beta$ at each step. Notably, the estimator for SGD in Lemma 3.1 is a special case of this formulation, obtained by setting $\beta = 0$, since the key distinction between SGD and SGD-M lies in the inclusion of momentum. The detailed proof is provided in the Appendix F.

The Adam optimizer (Kingma & Ba, 2015) uses adaptive moment estimates to adjust the learning rate for each parameter individually, resulting in faster convergence and improved performance. Given the two hyperparameters, $\beta_1$ and $\beta_2$, in Adam, we reformulate the parameter update in the form of SGD-M: $\theta_{t+1} = \theta_t - \eta_t^* \left( (1 - \beta_1) G^t + \beta_1 M_{t-1} \right)$. The ***general learning rate*** $\eta^*$ is a vector that $\eta^* = \eta_t / \left( (1 - \beta_1)(\sqrt{\hat{V}_t} + \epsilon) \right)$, where $\hat{V}_t = G_t^{\otimes 2} + \frac{\beta_2}{1-\beta_2} V_{t-1}$, $G_t^{\otimes 2}$ is the element-wise squaring operation on $G_t$. The vector $V$ could be easily obtained from the Adam optimizer in PyTorch (Adam Paszke et al., 2017). Thus, our estimator in Lemma 3.1 by just calculating $\alpha_k^t = \frac{1}{N} \left( \prod_{k<a<t} \eta_a^* \beta_1 \right) \eta_k^* (1 - \beta_1)$ with the ***general learning rate*** $\eta^*$.

## C  RELATED WORKS

### C.1  INFLUENCE ANALYSIS

Influence analysis is a technique to elucidate the connection between training data and model predictions (Hammoudeh & Lowd, 2022; Cook, 1986; Koh & Liang, 2017; Shapley et al., 1953; Chau et al., 2022). A straightforward way to measure the significance of a sample is to perform leave-one-out (LOO) retraining, which involves re-training the model on the training set without the sample and then comparing the discrepancy between it and the model trained on the full set. Nevertheless, the high computational expense of retraining for each sample is generally prohibitive due to the enormous size of contemporary deep-learning datasets (Russakovsky et al., 2015; Christoph Schuhmann et al., 2022; Soravit Changpinyo et al., 2021). Even some works that try to reduce the cost of retraining, such as by dividing the full set into subsets (Feldman & Chiyuan, 2020), still incur a high overall expense.

Instead of performing any retraining, Koh et al. (Koh & Liang, 2017) estimate the model change from a small weight change of training data. Specifically, this method utilizes the inverse-Hessian-gradient-product-based estimator to approximate the sample-wise influence. However, Koh's estimator has some limitations. The first one is that it relies on an overly strong assumption that the loss function with parameters should be convex, which is often not the practice case (Choromanska et al., 2015; Dauphin et al., 2014). The second limitation is that it can not be easily scaled up to large models and large datasets due to the heavy computation of the inverse-Hessian-gradient-product. The third shortcoming is that it cannot model the training dynamics since it only uses the parameters checkpoint from the very last iteration. Even so, it has attracted considerable attention from the academic community. Subsequently, there have been various research efforts to improve this method in various directions. For better scalability, a line of work (Grosse et al., 2023; Schioppa et al., 2022; Kwon et al., 2023) has been proposed. Grosse et al. (2023) proposed an efficient decomposition of Hessian to speed up the estimator's calculation. Another excellent work (Schioppa et al., 2022) introduced the novel Arnoldi iteration technique for accelerating the computation of the inverse Hessian, enabling applications to large-scale Transformer models in language and vision tasks, even when they have hundreds of millions of parameters. DataInf (Kwon et al., 2023) proposed a novel closed-form approximation for the inverse-Hessian better efficiency in both computational and memory complexities. Klochkov & Liu (2024) studied some hyper-parameters affecting the precision of using LiSSA algorithm to calculate the Hessian-related operation for influence function. Yang et al. (2024) proposed a Hessian-free approach to estimate the influence function by only calculating the inner product in the gradient space thereby achieving better scalability. As for the group effect (Yang et al., 2023), some works also analyze (Koh et al., 2019) and improve (Basu et al., 2020) the influence function on measuring group effects, for instance, Basu et al. (2020) extended influence functions to directly account for sub-population-group effects by considering higher-order terms in Taylor approximation. Moreover, PBRF (Bae et al., 2022) analyzes several practical reasons for the failure of Koh's estimator, *e.g.* the distinction between cold-start and warm-start response, the implicit regularizer, and the non-converged parameters. Additionally, it proposes the proximal Bregman response function to improve the performance. However, the requirement of Koh's estimator on the convex property of loss function and the neglect of training dynamics are not well solved in the above method. Recently, some work has attempted to estimate the effect of a sample by comparing the changes in the representation of some samples before and after additional training (Kim et al., 2024; Ko et al., 2024). They will perform better than Koh's estimator, but additional training will undoubtedly bring additional computational costs, especially if the data set is large. Chhabra et al. (2024b) proposed to combine the influence function with tree-structure to provide interpretations of which sample features contribute positively or negatively to the model's performance.

In recent years, a series of works represented by SGD-Inf (Hara et al., 2019), TracIn (Pruthi et al., 2020) and MoSo (Tan et al., 2023) have brought new solutions to this problem. All three are achieved by calculating the gradient information during training. For example, SGD-Inf (Hara et al., 2019) proposes an approximator with theoretical guarantees by tracking the gradient information of each sample at each epoch during training. TracIn (Pruthi et al., 2020) greatly improves the computational efficiency by sampling time steps. These schemes not only eliminate the requirement of convexity of the loss function in the calculation of influence but also fully perceive training dynamics. However, these methods are also problematic, they either do not estimate the influence of a sample on model parameters. In addition, in training, the choice of optimizer also has a significant impact on training,

---

**Algorithm 1:** Calculate differential influence function for a single sample $z$.

1: **Input:** A set of training data $\mathbf{D}$, a validation set $\mathbf{V}$, and a training sample $z \in \mathbf{D}$;
2: **Input:** Several checkpoints $(\theta^t)_{t \in \{t_1, \ldots, t_m\}}$ during training;
3: **Initialization:** $\mathcal{I}_\theta = \mathbf{0}$ and $\mathcal{I}(z, \mathbf{V}) = 0$;
4: **for** $t \in \{t_1, \ldots, t_m\}$ **do**
5:     Compute the influence difference $\mathcal{D}^t(z)$ by Eq.5;
6:     $\mathcal{I}_\theta \leftarrow \mathcal{I}_\theta + \mathcal{D}^t(z), \quad \mathcal{I}(z, \mathbf{V}) \leftarrow \mathcal{I}(z, \mathbf{V}) + \nabla\mathcal{L}(\mathbf{V}, \theta^t)^{\mathrm{T}} \mathcal{D}^t(z)$, see Eq.7;
7: **end for**
8: **Output:** $\mathcal{I}_\theta$ and $\mathcal{I}(z, \mathbf{V})$.

---

and these schemes are designed for one particular optimizer, namely the stochastic gradient descent (SGD). So they don't account for the impact of the optimizer's variability (*e.g.* the widely used Adam (Kingma & Ba, 2015)), thereby limiting their broader applicability.

Besides the aforementioned methods, another very effective scheme for estimating the influence is the Shapley value (Shapley et al., 1953) based on cooperative game theory. It can be interpreted as the contribution of each sample to the model's performance by quantifying the marginal gain in performance when a sample is added to a randomly selected subset. However, the computation of the Shapley value is very expensive, since it requires evaluating all possible subsets. Therefore, some approximation methods, such as Monte-Carlo sampling (Ghorbani & Zou, 2019), kernel method (Chau et al., 2022), or KNN-based method (Jia et al., 2019), have been proposed to reduce the computational cost.

### C.2 APPLICATIONS

The technique of influence analysis, which measures the impact of training samples on the model's performance, has been explored by the academic community in many scenarios. Here we list some representative topics and works. (1) Dataset pruning / Coreset selection (Yang et al., 2023; Jia et al., 2021; Choe et al., 2024): selecting a subset of the dataset that preserves the model's accuracy while reducing the size or complexity of the data. For example, LoGra (Choe et al., 2024) proposed an extremely efficient influence-based pipeline to conduct data valuation and selection for large language models. (2) Noise and outlier sample detection (Koh & Liang, 2017; Jia et al., 2021; Tang et al., 2021; Hara et al., 2019; Chhabra et al., 2024a) and noise label correction (Kong et al., 2022): identifying and removing or correcting the samples that have incorrect or misleading labels or features that degrade the model's quality. (3) Adversarial attack (Koh & Liang, 2017; Fang et al., 2020): generating samples that can fool or attack the model by exploiting its weaknesses or vulnerabilities. (4) Continual learning (Sun et al., 2022): the process of constantly monitoring and retraining machine learning models with updated data to prevent concept drifts and maintain accuracy and reliability. (5) Machine unlearning (Fu et al., 2022): removing the influence of a specific sample or group of samples from the model, which can be useful for privacy, security, or legal reasons. (6) Data attribution (Grosse et al., 2023; Chen et al., 2022a; Park et al., 2023; Dai & Gifford, 2023; Ilyas et al., 2022): attributing the model's output or behavior to the input data or features that contributed to it. For example, Li & Liu (2022) proposed an influence-based data reweighting pipeline to enhance better fairness. HYDRA (Chen et al., 2022b) attributes the model's output by unrolling the hyper-gradient of test loss throughout the training trajectory. Datamodels (Ilyas et al., 2022) proposed an interpretable pipeline by introducing a simple surrogate model (like a linear model) to understand the relation between data and prediction and then give rise to various interesting applications. (7) ISAL (Liu et al., 2021b) designed an active learning pipeline by utilizing the influence function to pick up the most influential data points at each iteration.

Table 7: Generalization to unseen architecture test on CIFAR-100 and Tiny-ImageNet (denoted by Tiny in the table). Here, we tested the generalization performance of the coresets selected using various methods on network architectures different from the surrogate network's architecture. We chose two different architectures, SENet (Jie Hu et al., 2018) and EfficientNet-B0 (Mingxing Tan & Quoc Le, 2019) (denoted by EB0 in the table). In all experiments, the selection ratio of the coresets from all methods is 20%.

| Settings | SENet CIFAR | EB0 CIFAR | SENet Tiny | EB0 Tiny |
|----------|-------------|-----------|------------|----------|
| Random   | 53.57       | 42.42     | 34.13      | 32.88    |
| SSP      | 54.16       | 43.65     | 31.74      | 30.99    |
| Moderate | 55.57       | 48.58     | 36.04      | 34.26    |
| IF       | 47.81       | 36.94     | 32.82      | 30.75    |
| DataInf  | 48.98       | 39.42     | 33.24      | 31.63    |
| TracIn   | 53.12       | 46.09     | 34.25      | 33.54    |
| GEX      | 51.67       | 42.87     | 34.84      | 31.93    |
| Ours     | **55.62**   | **48.82** | **36.41**  | **34.52**|

# D  SUPPLEMENTARY EXPERIMENTS

## D.1  GENERALIZATION TEST OF THE CORESET

To test whether the selected coresets are overfitting to the specific network architecture, we assessed their ability to generalize to different architectures using CIFAR-100 and Tiny-ImageNet datasets in our experiments. Specifically, we evaluated the performance of the coresets on two different architectures: SENet (Jie Hu et al., 2018) and EfficientNet-B0 (Mingxing Tan & Quoc Le, 2019), which were not used in the initial selection process.

Based on the results presented in Table 7, our approach consistently outperforms other methods in terms of generalization to unseen architectures. Specifically, our method achieves the highest performance on both SENet and EfficientNet-B0 architectures for both CIFAR-100 and Tiny-ImageNet datasets. For example, on the EfficientNet-B0 architecture over CIFAR-100, our method achieves an accuracy of 48.82%, which is 0.24% higher than the second-best method (Moderate). On the Tiny-ImageNet dataset, our method again achieves the highest accuracy of 36.41% and 34.52% on the SENet and EfficientNet-B0 architectures, respectively. It is worth noting that some methods, such as IF, DataInf, and GEX, sometimes perform even worse than the randomly selected random coreset, while our Diff-In achieves the largest margin of improvement over random selection. This demonstrates the good generalization ability of our method.

## D.2  FURTHER STUDY ON DATA CLEANING

Here, we conducted further experiments on data cleaning as a supplement to the experiments in Section 5.1 of the main text. In this set of experiments, all data-cleaning techniques are applied to the Tiny-ImageNet (Ya Le & Xuan S. Yang, 2015) dataset. The purpose of this experiment is to simulate the noise that can occur during the annotation process by introducing a certain percentage of random label noise, denoted as $r_n\%$. The experiment aims to evaluate the ability of different methods to identify and remove noisy data from the dataset.

The results are presented in Table 8, where the numbers represent the proportion of noise samples in the selected $r_n\%$ data points to the total noise samples in the entire dataset. A higher number in the table indicates a better ability of the method to identify and filter out noisy data. Our method outperforms others by a significant margin. Note that as the noise ratio increases, the performance of all methods except our method gradually decreases to the level of random selection, while our method's performance remains stable.

We visually displayed some results of identifying noisy samples in Figure 4. We compared the results of identifying noise by IF (Koh & Liang, 2017) and our Diff-In. In this experiment, 20% of the data labels were randomly replaced with incorrect labels to introduce label noise. Diff-In is a powerful

Table 8: Additional data cleaning experiment on Tiny-ImageNet (Ya Le & Xuan S. Yang, 2015). For all experiments, we introduce $r_n\%$ random label noise to simulate the noise that can occur during the annotation process. The numbers in the table below represent the proportion of noise samples in the selected data with a ratio of $r_n\%$ to the total noise samples in the entire data set. The larger the number, the better the method's ability to identify noise data.

| Dataset | Tiny-ImageNet | | | | | | | |
|---|---|---|---|---|---|---|---|---|
| Noise ratio $r_n\%$ | 10% | 20% | 30% | 40% | 50% | 60% | 70% | 80% |
| Selection ratio | 10% | 20% | 30% | 40% | 50% | 60% | 70% | 80% |
| Random | 10.0 | 20.0 | 30.0 | 40.0 | 50.0 | 60.0 | 70.0 | 80.0 |
| Entropy | 11.1 | 21.2 | 31.6 | 41.2 | 50.9 | 60.7 | 70.2 | 80.1 |
| Margin | 10.7 | 20.6 | 30.7 | 40.7 | 50.5 | 60.3 | 70.1 | 80.1 |
| IF(Koh & Liang, 2017) | 10.4 | 21.1 | 36.1 | 40.7 | 50.1 | 60.0 | 69.9 | 79.9 |
| DataInf(Kwon et al., 2023) | 12.5 | 26.9 | 37.6 | 41.3 | 53.1 | 62.3 | 71.5 | 80.5 |
| TracIN(Pruthi et al., 2020) | 70.7 | 76.3 | 79.0 | 76.7 | 75.7 | 70.4 | 70.2 | 82.3 |
| GEX(Kim et al., 2024) | 65.9 | 72.7 | 74.7 | 75.0 | 72.4 | 71.6 | 71.5 | 80.5 |
| Ours | **78.6** | **88.2** | **85.4** | **86.3** | **88.1** | **84.8** | **88.9** | **87.5** |

noise-label indicator, while IF identifies more difficult samples that are correct in category but have some recognition difficulty in image content.

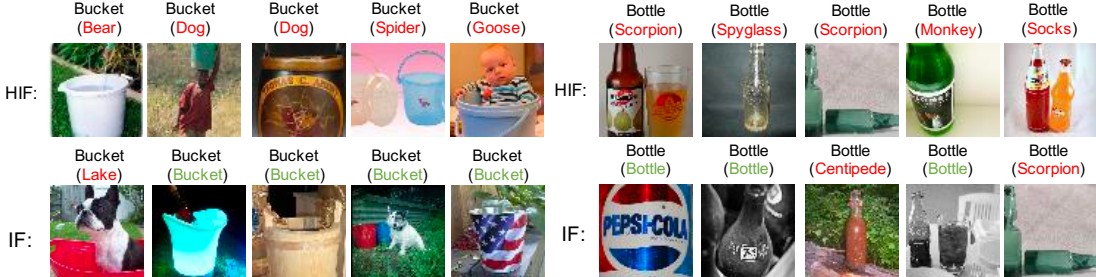

Figure 4: Some results of data cleaning results by ours and IF (Koh & Liang, 2017). In this experiment, 20% of the data labels were replaced with random incorrect labels to introduce label noise. Each image in the above corresponds to a ground-truth label in black and a generated label, which is represented by a wrong label in red or a correct label in green.

## D.3 Experimental Results with STD

Due to limited space in the main paper, we present experimental results with std information here as a supplement to Table 3 and Table 1. Please see Table 9 for results of the coreset experiment with std information. And please check Table 10 for the results of the data cleaning with std information.

Table 9: Experimental results with std-values of coreset selection on three public datasets with ResNet-50 model. Training on the full CIFAR-100, Tiny-ImageNet, and ImageNet-1K datasets without data selection achieved 78.1%, 49.3%, and 76.4%, respectively. The best results are bolded for baselines and ours, respectively.

| Dataset | CIFAR-100 | | | | | Tiny-ImageNet | | | | | ImageNet-1K | | | | |
|---|---|---|---|---|---|---|---|---|---|---|---|---|---|---|---|
| Selection Rate | 20% | 30% | 50% | 70% | 80% | 20% | 30% | 50% | 70% | 80% | 20% | 30% | 40% | 60% | 80% |
| Random | 50.2 | 53.6 | 64.3 | 71.0 | 74.1 | 24.0 | 29.7 | 34.4 | 40.9 | 45.7 | 61.6 | 65.9 | 67.7 | 70.3 | 72.9 |
| SSP (Sorscher et al., 2022) | 44.4$_{\pm2.5}$ | 54.6$_{\pm2.1}$ | 62.9$_{\pm1.2}$ | 70.7$_{\pm0.8}$ | 75.2$_{\pm0.4}$ | 20.8$_{\pm0.4}$ | 27.6$_{\pm0.5}$ | 32.5$_{\pm0.3}$ | 39.6$_{\pm0.3}$ | 44.9$_{\pm0.3}$ | 31.2$_{\pm0.6}$ | 51.4$_{\pm0.6}$ | 60.3$_{\pm0.5}$ | 69.8$_{\pm0.6}$ | **75.5**$_{\pm0.4}$ |
| Moderate (Xia et al., 2023) | 51.8$_{\pm1.5}$ | 57.7$_{\pm1.6}$ | 64.9$_{\pm0.9}$ | **71.8**$_{\pm0.8}$ | 74.2$_{\pm0.4}$ | 25.2$_{\pm0.3}$ | **30.5**$_{\pm0.2}$ | 34.8$_{\pm0.5}$ | **41.4**$_{\pm0.4}$ | 46.0$_{\pm0.3}$ | 61.1$_{\pm0.6}$ | 67.8$_{\pm0.4}$ | **70.0**$_{\pm0.5}$ | 73.0$_{\pm0.4}$ | 75.3$_{\pm0.4}$ |
| IF (Koh & Liang, 2017) | 26.4$_{\pm1.3}$ | 36.1$_{\pm1.2}$ | 47.1$_{\pm1.1}$ | 51.2$_{\pm0.8}$ | 63.5$_{\pm0.6}$ | 14.3$_{\pm1.2}$ | 19.1$_{\pm1.1}$ | 24.5$_{\pm1.0}$ | 31.2$_{\pm0.8}$ | 37.9$_{\pm0.6}$ | 25.6$_{\pm1.1}$ | 30.5$_{\pm1.0}$ | 49.9$_{\pm0.9}$ | 56.7$_{\pm0.7}$ | 68.1$_{\pm0.6}$ |
| DataInf (Kwon et al., 2023) | 26.9$_{\pm1.6}$ | 35.8$_{\pm1.3}$ | 48.2$_{\pm1.1}$ | 52.5$_{\pm0.9}$ | 65.5$_{\pm0.7}$ | 16.1$_{\pm1.3}$ | 21.2$_{\pm1.2}$ | 27.7$_{\pm1.0}$ | 33.1$_{\pm0.9}$ | 42.4$_{\pm0.7}$ | 27.7$_{\pm1.2}$ | 32.3$_{\pm1.1}$ | 52.6$_{\pm1.0}$ | 58.2$_{\pm0.8}$ | 67.0$_{\pm0.8}$ |
| TracIn (Pruthi et al., 2020) | 50.4$_{\pm1.0}$ | 58.4$_{\pm0.9}$ | 63.9$_{\pm0.8}$ | 70.4$_{\pm0.6}$ | 72.3$_{\pm0.6}$ | 24.2$_{\pm0.7}$ | 28.8$_{\pm0.6}$ | 35.3$_{\pm0.6}$ | 40.1$_{\pm0.5}$ | 45.8$_{\pm0.4}$ | 61.3$_{\pm0.7}$ | 64.4$_{\pm0.6}$ | 70.3$_{\pm0.6}$ | 71.4$_{\pm0.5}$ | 73.6$_{\pm0.5}$ |
| GEX (Kim et al., 2024) | 44.8$_{\pm1.1}$ | 50.0$_{\pm0.9}$ | 57.5$_{\pm0.8}$ | 62.0$_{\pm0.6}$ | 69.9$_{\pm0.6}$ | 20.5$_{\pm0.8}$ | 23.0$_{\pm0.8}$ | 29.4$_{\pm0.8}$ | 36.6$_{\pm0.8}$ | 41.6$_{\pm0.6}$ | 55.3$_{\pm0.8}$ | 58.7$_{\pm0.7}$ | 65.7$_{\pm0.6}$ | 69.8$_{\pm0.6}$ | 70.1$_{\pm0.4}$ |
| Ours | **52.0**$_{\pm1.0}$ | **59.4**$_{\pm1.0}$ | **65.7**$_{\pm0.8}$ | 71.5$_{\pm0.7}$ | **75.3**$_{\pm0.6}$ | **26.6**$_{\pm0.8}$ | **30.5**$_{\pm0.6}$ | **36.0**$_{\pm0.5}$ | 40.9$_{\pm0.4}$ | **46.4**$_{\pm0.4}$ | **61.7**$_{\pm0.6}$ | **68.3**$_{\pm0.6}$ | 69.8$_{\pm0.5}$ | **73.6**$_{\pm0.6}$ | **75.5**$_{\pm0.4}$ |

Table 10: Experimental results with std values of data cleaning, aiming to identify the noisy data of the given dataset. The reported performance represents the percentage of the noisy samples contained in the selected subset (under different selection rates) compared to the total number of noisy samples in the entire dataset. The best results are bolded.

| Dataset | SVHN | | | | | Tiny-ImageNet | | | | | GSM8K | | | | |
|---|---|---|---|---|---|---|---|---|---|---|---|---|---|---|---|
| Selection Rate | 20% | 30% | 40% | 50% | 60% | 20% | 30% | 40% | 50% | 60% | 20% | 30% | 40% | 50% | 60% |
| Random | 20.0 | 30.0 | 40.0 | 50.0 | 60.0 | 20.0 | 30.0 | 40.0 | 50.0 | 60.0 | 20.0 | 30.0 | 40.0 | 50.0 | 60.0 |
| Loss value | $27.2_{\pm3.1}$ | $43.4_{\pm3.4}$ | $51.0_{\pm3.3}$ | $65.9_{\pm2.9}$ | $69.1_{\pm2.9}$ | $21.2_{\pm3.1}$ | $31.7_{\pm2.8}$ | $42.1_{\pm2.1}$ | $52.2_{\pm2.1}$ | $62.6_{\pm1.7}$ | $28.4_{\pm2.4}$ | $39.3_{\pm2.2}$ | $51.5_{\pm1.9}$ | $57.4_{\pm1.8}$ | $63.7_{\pm2.0}$ |
| IF (Koh & Liang, 2017) | $41.6_{\pm1.8}$ | $46.3_{\pm1.7}$ | $59.5_{\pm1.5}$ | $66.8_{\pm1.3}$ | $72.5_{\pm1.3}$ | $21.1_{\pm1.4}$ | $31.6_{\pm1.5}$ | $42.0_{\pm1.4}$ | $52.2_{\pm1.1}$ | $62.3_{\pm1.1}$ | $67.2_{\pm3.2}$ | $71.4_{\pm2.5}$ | $79.6_{\pm2.1}$ | $88.5_{\pm2.4}$ | $94.7_{\pm1.4}$ |
| DataInf (Kwon et al., 2023) | $42.9_{\pm2.1}$ | $46.6_{\pm1.9}$ | $61.9_{\pm1.6}$ | $68.5_{\pm1.6}$ | $72.9_{\pm1.5}$ | $26.9_{\pm2.3}$ | $38.6_{\pm2.2}$ | $45.5_{\pm2.2}$ | $58.6_{\pm1.8}$ | $64.8_{\pm1.7}$ | $68.5_{\pm1.7}$ | $73.4_{\pm1.4}$ | $84.2_{\pm1.6}$ | $90.1_{\pm1.7}$ | $96.2_{\pm1.1}$ |
| TracIN (Pruthi et al., 2020) | $81.4_{\pm1.7}$ | $90.2_{\pm1.1}$ | $95.5_{\pm0.5}$ | $98.2_{\pm0.1}$ | $99.2_{\pm0.1}$ | $76.3_{\pm2.4}$ | $91.0_{\pm0.7}$ | $95.9_{\pm0.7}$ | $97.8_{\pm0.3}$ | $98.4_{\pm0.3}$ | $78.4_{\pm2.5}$ | $84.5_{\pm1.1}$ | $92.2_{\pm0.2}$ | $99.2_{\pm0.2}$ | $99.2_{\pm0.2}$ |
| GEX (Kim et al., 2024) | $80.7_{\pm2.2}$ | $84.9_{\pm1.9}$ | $86.7_{\pm1.7}$ | $89.5_{\pm1.5}$ | $91.5_{\pm1.2}$ | $72.7_{\pm1.9}$ | $77.2_{\pm1.8}$ | $81.9_{\pm1.8}$ | $87.3_{\pm1.7}$ | $90.1_{\pm1.4}$ | $77.2_{\pm1.8}$ | $87.1_{\pm1.4}$ | $91.6_{\pm1.3}$ | $95.3_{\pm0.9}$ | $99.9_{\pm0.0}$ |
| Ours | $\mathbf{90.2}_{\pm0.7}$ | $\mathbf{96.4}_{\pm0.6}$ | $\mathbf{98.7}_{\pm0.4}$ | $\mathbf{99.6}_{\pm0.1}$ | $\mathbf{99.6}_{\pm0.2}$ | $\mathbf{88.2}_{\pm1.6}$ | $\mathbf{98.0}_{\pm0.4}$ | $\mathbf{98.6}_{\pm0.4}$ | $\mathbf{99.0}_{\pm0.2}$ | $\mathbf{99.8}_{\pm0.0}$ | $\mathbf{86.1}_{\pm1.7}$ | $\mathbf{92.2}_{\pm0.8}$ | $\mathbf{99.3}_{\pm0.2}$ | $\mathbf{99.9}_{\pm0.1}$ | $\mathbf{99.9}_{\pm0.1}$ |

Table 11: Further experimental results on the influence of the choice of hyper-parameter $m$ on the performance and speed. The experiments are coreset selection for the supervised fine-tuning of a Llama-3-8B-instruct model on the OpenMathInstruct dataset. We report the performance on GSM8K of supervised fine-tuning large language models on the selected coreset. The selection rate is 30%.

| Method | 8-shot (CoT) on GSM8K | Time cost of the selection procedure (GPU-hours) |
|---|---|---|
| The Llama model | 77.2 | - |
| SFT on all data | 85.3 | - |
| DataInf (Kwon et al., 2023) | 79.8 | 15.2 |
| EK-FAC (Grosse et al., 2023) | 80.3 | 18.4 |
| Arnoldi(Schioppa et al., 2022) | 80.4 | 19.9 |
| TraK (Park et al., 2023) | 82.7 | 21.7 |
| TracIn (m=5) (Pruthi et al., 2020) | 81.3 | 26.3 |
| Diff-In (m=3) | 82.1±1.03 | 17.4 |
| Diff-In (m=5) | 84.4±0.39 | 28.7 |
| Diff-In (m=10) | 84.6±0.11 | 56.8 |

## D.4 FURTHER EXPERIMENTS ON LARGE LANGUAGE MODELS

Here, we show more studies on the choice of the hyper-parameter $m$ (which is the number of selected time steps in the calculation of Diff-In), comparison with SOTA methods, and the guidelines for the selection of random time steps. The first set of results is on the coreset selection experiment for the supervised fine-tuning of a Llama-3-8B-instruct model on the OpenMathInstruct dataset (Toshniwal et al., 2024) (a dataset containing 1.8M high-quality question-answer pairs). The selection pipeline is: first LoRA fine-tuning the model on all data, then estimation sample-wise influence with the LoRA parameters by different influence estimators, and finally selecting the most influential samples as the coreset.

### D.4.1 FURTHER PERFORMANCE AND SPEED COMPARISON WITH SOTA METHODS

The selected baselines contain: TracIn (Pruthi et al., 2020), TraK (Park et al., 2023), DataInf (Kwon et al., 2023), and EK-FAC (Grosse et al., 2023), and Arnoldi-IF (Schioppa et al., 2022). The results are shown in Table 11. Diff-In (m=5) outperformed TracIn and TRAK by 3.1 Acc and 1.7 Acc respectively, at the cost of an extra 2 hours compared with TracIn. EK-FAC is the fastest, but it is also significantly behind in performance (because it is still based on the convexity assumption for the deep models). Overall, the computation time across all methods (m=3 or 5 for Diff-In) was comparable, efficient, and scalable to large-scale models. The results demonstrate that Diff-In outperforms all other methods, highlighting its effectiveness in measuring sample importance for large datasets and models.

Moreover, we also conduct Diff-In with two influence-based effective outlier and noise detection approaches, namely IP (Yang et al., 2024) and OGI (Chhabra et al., 2024a). As for introducing these two baselines, we recommend Sec.C.1 and Sec.C.2. This experiment is a supplement to the GSM8K cleaning experiments in Sec.5.1, and we keep the basic settings all the same. The noise rate is 20%. Experimental results are shown in Table 12. In these experiments, Diff-In consistently achieved significantly superior performance, further demonstrating its effectiveness. Moreover, we also implement the Diff-In for the validation influence for data cleaning. A slight drop in performance was observed, suggesting that self-influence may be a more effective method for identifying incorrect or outlier data. Additionally, self-influence is computationally more efficient, as it requires only the sample itself to calculate influence, rather than relying on a separate validation dataset.

Table 12: Comparison of the noise sample cleaning experiments for GSM8K as a supplement to the GSM8K cleaning experiments in Sec.5.1. The newly selected baselines contain IP (Koh & Liang, 2017), OGI (Chhabra et al., 2024a). The noise rate is 20%.

| Method | Precision at the selection rate of 20% |
|---|---|
| IF | 67.2 |
| DataInf (Kwon et al., 2023) | 68.5 |
| TracIn (Grosse et al., 2023) | 78.4 |
| Gex(Schioppa et al., 2022) | 77.2 |
| IP (Park et al., 2023) | 74.4 |
| OGI (Pruthi et al., 2020) | 81.2 |
| Diff-In (self-influence) (m=5) | 86.1 |
| Diff-In (validation-influence) (m=5) | 85.8 |

Table 13: Furthe ablation study for the settings of the number of selected time steps on CC12M.

| The choice of $m$ | Zero - shot classification on ImageNet - 1K | Linear Prob on ImageNet - 1K | I2T Retrieval | T2I Retrieval |
|---|---|---|---|---|
| 3 | $24.3 \pm 1.25$ | $33.2 \pm 0.42$ | $20.7 \pm 1.14$ | $46.5 \pm 1.21$ |
| 5 | $25.8 \pm 0.43$ | $34.4 \pm 0.10$ | $23.9 \pm 0.82$ | $48.3 \pm 0.79$ |
| 20 | $26.1 \pm 0.21$ | $34.6 \pm 0.11$ | $24.4 \pm 0.65$ | $48.2 \pm 0.31$ |

### D.4.2 THE EFFECT OF THE CHOICE OF $m$

This study was performed on two large-scale scenarios: vision-language pre-training (VLP) using CLIP-ViT-Base on CC12M (Soravit Changpinyo et al., 2021) and supervised fine-tuning (SFT) of a large language model on the OpenMathInstruct dataset (Toshniwal et al., 2024). First, we present the VLP results at a selection rate of 10% in Table 13. The results indicate that as increases, both performance and stability improve. However, this improvement plateaus once exceeds 5.

Then, on the OpenMathInstruct, we change the choice of m from 3 to 10 for our DIff-In to study the influence of the choice of the number $m$ of selected checkpoints on the performance and speed. We show the results in Table 11. The results show the same trend as that in the ablation study in Sec.5.4. Specifically, Diff-In consistently delivers the best performance. As $m$ increases, both performance and stability improve, with diminishing returns once $\geqslant 5$. These findings align with the ablation study in the main paper. We observed that with 5 checkpoints, Diff-In incurred only about 2 additional GPU hours compared to TracIn while improving performance by over 3%. This checkpoint configuration is robust across different settings and does not require re-validation on new datasets and tasks.

All these conclusions are consistent with that from the ablation study on CIFAR in Sec.5.4.

### D.4.3 GUIDELINES FOR CHECKPOINT SELECTION

In addition, we found that the effect of random checkpoint selection is good and stable. The intrinsic reason for this is that the uniformly random sample can guarantee that the selected checkpoint will span uniformly across different training stages. In the middle and later stages of model training, the model changes very little. This also reveals the guideline for selecting checkpoints, that is, to sample evenly and ensure that the time step is selected across different training stages. To further illustrate this, we set up three sampling modes based on the coreset experiments of ResNet-50 and ImageNet-1K at a selection rate of 20%. The mode-1 is to randomly sample 5 checkpoints, mode-2 is to select the checkpoints of the first 5 epochs in the training, and mode-3 is to select the checkpoints of the last 5 epochs. The final results of these three modes are: 61.7, 54.3, 60.2. We found that the approximation accuracy of mode 1 was significantly better!

### D.5 FURTHER EXPERIMENTS AND DISCUSSION FOR THE BOUND OF DIFF-IN

### D.5.1 POLYNOMIAL ERROR BOUND

The bound proposed in Theorem 4.1 shows that the upper bound of the approximation error of Diff-In grows with the increase of the training times $T$. The reason that increasing the training steps leads to larger errors is that the optimized parameters diverge further from the initialized parameters as training progresses. This divergence makes accurate estimation more challenging and contributes to

the accumulation of approximation errors. It is also worth noting that some methods, such as those in (Hara et al., 2019; Schioppa et al., 2024), exhibit a faster increase in error as $T$ grows. In contrast, our approach, with its polynomial error bound, demonstrates a significantly smaller growth rate compared to other methods, such as the exponential growth observed with SGD-Inf (Hara et al., 2019). This indicates that our method effectively mitigates error accumulation over time, even as $T$ increases.

It is important to emphasize that this bound reflects a worst-case scenario for error. In practice, Diff-In performs robustly even with larger $T$ values. As training progresses and the model approaches convergence, parameter changes become minimal, meaning larger $T$ typically has a negligible impact on errors. Here, we show the performance change of the coreset selected by Diff-In vs the change of T on ImageNet. When $T = 50$ (the default setting in this paper), the top-1 accuracy of the model trained on coreset is 61.7. When $T = 100$ and $T = 200$, the performance is 61.4 and 62.4 respectively. Notably, increasing $T$ does not degrade Diff-In's practical performance.

### D.5.2 BASIC ASSUMPTIONS ON THE SMOOTHNESS

Lipschitz continuity and gradient norms are commonly used to characterize the smoothness of a neural network's loss landscape. Modern deep learning models incorporate techniques such as normalization layers (e.g., batch normalization and layer normalization) and shortcut connections to enhance smoothness and continuity (Santurkar et al., 2018; Li et al., 2018), facilitating optimization. These techniques make the assumption of smoothness generally valid in practice. This stands in contrast to many studies (Koh & Liang, 2017; Grosse et al., 2023) that rely heavily on the convexity of neural networks. Our assumption, by comparison, is much easier to satisfy in real-world scenarios. If the conditions for $\ell$-Lipschitz continuous gradients are not met, the gradient norms $g$, the values of $g$ and $\ell$ can become very large. In such cases, the error bound derived from these parameters may lose its practical relevance. However, this does not necessarily imply that the algorithm will fail in practice.

## E EXPERIMENTAL SETTINGS

### E.1 DATASETS AND GENERAL SETTINGS

Here, we introduce the datasets used in this work: (1). CIFAR-100 (Alex Krizhevsky, 2009) consists of 50,000 training examples categorized into 100 classes. (2). CIFAR-10 (Alex Krizhevsky, 2009) has 50,000 training images and 10,000 test images belonging to 10 different classes. (3). Tiny-ImageNet (Ya Le & Xuan S. Yang, 2015) comprises 100,000 images belonging to 200 classes. (4). SVHN (Netzer et al., 2011), which stands for Street View House Numbers, is an image digit recognition dataset with 73,257 training images and 26,032 testing images. (5). ImageNet-1K (Russakovsky et al., 2015) covers 1,000 classes and contains over 1 million training images. (6). SM8K (Grade School Math 8K) is a dataset launched by OpenAI (Cobbe et al., 2021), aiming to study the capabilities of large language models in solving mathematical problems. This dataset contains 8500 high-quality and linguistically diverse primary school math problems, mainly involving basic arithmetic operations. Here we provide an example problem:

*Problem: Beth bakes 4.2 dozen batches of cookies in a week. If these cookies are shared equally among 16 people, how many cookies does each person consume?*

*Answer: 6*

Note that when approximating Diff-In, we need to compute the information over the entire dataset, *e.g.*, the gradient $G = \nabla\mathcal{L}(\mathcal{D}, \theta)$ over the whole dataset. In practice, we use a random batch as an efficient proxy to the entire dataset. This random batch has a size of $2048$ for experiments on ImageNet-1K (Russakovsky et al., 2015) while $512$ for others. Similar proxy schemes were also adopted in previous works (Koh & Liang, 2017; Tan et al., 2023; Pruthi et al., 2020; Yang et al., 2023).

### E.2 CORESET SELECTION.

Our experiments were mainly conducted with Pytorch (Adam Paszke et al., 2017) on a server with 8 Tesla-V100 GPUs. ResNet-50 (He et al., 2016) is adopted for surrogate network training and

final training on the coreset. All the surrogate networks are trained for 50 epochs for efficiency. For CIFAR-100, we utilize the SGD optimizer with a learning rate of 0.1 and a batch size of 128. For Tiny-ImageNet, we use the SGD optimizer with a learning rate of 0.3 and a batch size of 64. For ImageNet-1K, we use the SGD optimizer with a learning rate of 0.4 and a batch size of 256. For data augmentation, we only adopt RandomResizedCrop and RandomHorizontalFlip in the TorchVision package for all experiments. To calculate the mathematical expectation in Proposition 3.2, we randomly select 5 time steps. For the training on the final subset, we set the maximum epochs for CIFAR-100, Tiny-ImageNet, and ImageNet-1K as 200, 200, and 100 epochs, respectively. For coreset selection on the vision-language dataset CC12M (Soravit Changpinyo et al., 2021), all experiments are conducted on 16 NVIDIA V100 GPUs. The selected model is CLIP model (Alec Radford et al., 2021). We follow the settings provided in the original paper. Specifically, the CLIP model Alec Radford et al. (2021) is trained for 32 epochs with AdamW optimizer, weight decay 0.2, and a batch size of 2048. After 1 warmup epoch, the learning rate gradually decreases from 1e-4 following the cosine strategy.

**Zero-shot ImageNet classification.** The CLIP model has two encoders, one for text and one for images. During the zero-shot classification process, text descriptions corresponding to the ImageNet classes are formulated. For example, if a class "dog" exists, a text description like "a picture of a dog" might be created. These text descriptions are encoded by the text encoder of CLIP to obtain text embeddings. At the same time, the images from the ImageNet dataset are encoded by the image encoder of CLIP to get image embeddings. Then, the similarity between each image embedding and all the text embeddings (representing different classes) is calculated. The image is classified into the class whose text embedding has the highest similarity to the image embedding.

**Linear Prob.** This is a technique used to evaluate and analyze the performance of a pre-trained model. For the CLIP model, linear probing involves adding a linear layer on top of the pre-trained CLIP model and then training only this linear layer while keeping the rest of the CLIP model's parameters fixed.

**Image-Text Retrieval.** This is a task where the goal is to find the most relevant text description for a given image or find the most relevant image for a given text description. Let us use the Image-to-Text Retrieval as an example. The image is encoded using the vision encoder. This results in an image embedding that represents the visual features of the image. Then, text documents are also encoded (using the text encoder) to obtain their respective text embeddings. The similarity between the image embedding and all the text embeddings is computed. The text with the highest similarity score is retrieved as the relevant description for the image.

### E.3 DATA CLEANING AND DATA DELETION

For experiments on the (classification) data cleaning and data deletion experiments, we use ResNet-18 (He et al., 2016) is adopted for the three datasets, namely CIFAR-10/100 (Alex Krizhevsky, 2009) and Tiny-ImageNet (Ya Le & Xuan S. Yang, 2015). For experiments on CIFAR-10/100, we utilize the SGD optimizer with a learning rate of 0.1 and a batch size of 128, with the maximum epoch setting as 200. For experiments on Tiny-ImageNet, we utilize the SGD optimizer with a learning rate of 0.3 and a batch size of 128, with the maximum epoch setting as 200. To calculate the mathematical expectation in Proposition 3.2, we also randomly select 5 time steps as in the coreset experiments.

**For the data cleaning on GSM8K:** Here we provide an example problem from GSM8K:

*Problem: Beth bakes 4.2 dozen batches of cookies in a week. If these cookies are shared equally among 16 people, how many cookies does each person consume?*

*Answer: 6*

Here, in order to test the current various schemes' ability to identify noisy data, we introduce label noise by randomly perturbing the answer. For example, we will randomly change the answer in the above text to other numbers (for example, change the answer 6 to 9). Then, we will use the corrupted data to perform LoRA fine-tuning on the Llama 3.1 8B model. Finally, we calculate the self-influence of each sample during the LoRA fine-tuning process using each influence estimator, where the involved parameters are all LoRA parameters (Xia et al., 2024; Kwon et al., 2023). The specific settings for LoRA fine-tuning are as follows: the Lora-rank is 64, bf-16 precision is used, the number of epochs is 2, the Lora-target-modules include q-proj, k-proj, v-proj, o-proj, the learning

rate is $1e^{-05}$, the batch size is 8, the gradient accumulation steps is 16, and the AdamW optimizer is used. This experiments is conducted on a server with 8 A100 GPUs.

### E.4 APPROXIMATION PRECISION STUDY

In Figure 1, Figure 3(a), and Figure 3(b) from the main paper, we examined the precision of Diff-In in approximating the self-influence, which involves calculating $\mathcal{I}(z, \mathbf{V})$ with $\mathbf{V} = \{z\}$. We choose 30 data points with the highest influence scores for each type of influence estimator and then calculate the correlation between the estimated values and the exact value obtained by the brute-force LOO retraining. The model and dataset used for each experiment are ResNet-18 and CIFAR-10 for Figure 1(a), ResNet-101 and CIFAR-10 for Figure 1(b), ResNet-18 and ImageNet-1K for Figure 1(c), ResNet-18/Swin-Tiny and CIFAR-10 for Figure 3(a), ResNet-18 and CIFAR-10 for Figure 3(b) and (c).

## F    PROOF FOR PROPOSITION 3.2

**Proposition 3.2** *Let us keep the symbol convention of $\mathcal{D}^t(z)$, $c_t$ and $\alpha_t$ from Lemma 3.1. By using Lemma 3.1, the calculation for $\mathcal{I}_\theta(z)$ and $\mathcal{I}(z, \mathbf{V})$ could be formulated as:*

$$
\begin{aligned}
\mathcal{I}_\theta(z) &= T \cdot \mathbb{E}_{t \sim P(t)}\Big(\mathcal{D}^t(z)\Big), \\
\mathcal{I}(z, \mathbf{V}) &= T \cdot \mathbb{E}_{t \sim P(t)}\Big\langle G_\mathbf{V}^t, \mathcal{D}^t(z)\Big\rangle,
\end{aligned}
\tag{13}
$$

*where the $P(t) = Uniform(0, ..., T-1)$ is the uniform time-step distribution, $G_\mathbf{V}^t = \nabla \mathcal{L}(\mathbf{V}, \theta^t)$ indicates the gradient over the query set $\mathbf{V}$ of parameters $\theta^t$ at the $t-th$ iteration, and $\langle \cdot, \cdot \rangle$ is the inner product operation.*

**Proof.** First, we revisit the differentiation operation. Let $\mathcal{I}_\theta^t(z)$ denote the influence on parameters at the $t$-th iteration caused by just removing the sample $z$ from the training set, and $\mathcal{D}^t(z) = \mathcal{I}_\theta^{t+1}(z) - \mathcal{I}_\theta^t(z)$ denote the sample-wise influence difference between adjacent time steps. First, we differentiate the influence by representing it as the sum of the differences between all two adjacent time steps,

$$
\mathcal{I}_\theta(z) = \sum_{t < T} \mathcal{D}^t(z) + \mathcal{I}_\theta^0(z),
\tag{14}
$$

where $\mathcal{I}_\theta^0(z) = 0$ since there is no training for the initial guess. Lemma 3.1 offers an estimate for the difference in influence terms. Therefore, the main focus of the proof for Proposition 3.2 lies in demonstrating the validity of Lemma 3.1.

For the $\mathcal{I}(z, \mathbf{V})$, we can also differentiate it:

$$
\begin{aligned}
\mathcal{I}(z, \mathbf{V}) &= \sum_{t < T} \Big[\mathcal{I}^{t+1}(z, \mathbf{V}) - \mathcal{I}^t(z, \mathbf{V})\Big] \\
&= \sum_{t < T} \Big[\mathcal{L}(\mathbf{V}, \theta_{-z}^{t+1}) - \mathcal{L}(\mathbf{V}, \theta^{t+1}) - \mathcal{L}(\mathbf{V}, \theta_{-z}^t) + \mathcal{L}(\mathbf{V}, \theta^t)\Big].
\end{aligned}
\tag{15}
$$

By adopting the first-order Taylor approximation, we have

$$
\mathcal{L}(\mathbf{V}, \theta^{t+1}) - \mathcal{L}(\mathbf{V}, \theta^t) \approx G^t\Big[\theta^{t+1} - \theta^t\Big],
\tag{16}
$$

and

$$
\begin{aligned}
&\mathcal{L}(\mathbf{V}, \theta_{-z}^{t+1}) - \mathcal{L}(\mathbf{V}, \theta_{-z}^t) \\
={}&\mathcal{L}(\mathbf{V}, \theta_{-z}^{t+1}) - \mathcal{L}(\mathbf{V}, \theta^t) + \mathcal{L}(\mathbf{V}, \theta^t) - \mathcal{L}(\mathbf{V}, \theta_{-z}^t) \\
={}&\mathcal{L}(\mathbf{V}, \theta_{-z}^{t+1}) - \mathcal{L}(\mathbf{V}, \theta^t) - \Big(\mathcal{L}(\mathbf{V}, \theta_{-z}^t) - \mathcal{L}(\mathbf{V}, \theta^t)\Big).
\end{aligned}
\tag{17}
$$

If we use $\theta_1$ and $\theta_2$ to denote $\theta^t$ and $\theta_{-z}^{t+1}$ respectively, we have:

$$
L(\mathbf{V}, \theta_2) - L(\mathbf{V}, \theta_1) \approx \nabla L(\mathbf{V}, \theta_1)(\theta_2 - \theta_1),
$$

where that $\mathbf{V}$ is the validation set that has nothing with the training process. So, we have

$$
\begin{aligned}
&\mathcal{L}(\mathbf{V}, \theta_{-z}^{t+1}) - \mathcal{L}(\mathbf{V}, \theta_{-z}^t) \\
\approx{}&G_\mathbf{V}^t\Big[\theta_{-z}^{t+1} - \theta^t\Big] - G_\mathbf{V}^t\Big[\theta_{-z}^t - \theta^t\Big] \\
={}&G_\mathbf{V}^t\Big[\theta_{-z}^{t+1} - \theta_{-z}^t\Big].
\end{aligned}
\tag{18}
$$

Hence, the differentiation form of $\mathcal{I}(z, \mathbf{V})$ could be estimated as

$$
\begin{aligned}
&\mathcal{I}(z, \mathbf{V}) \\
={}&\sum_{t < T} \Big[\mathcal{L}(\mathbf{V}, \theta_{-z}^{t+1}) - \mathcal{L}(\mathbf{V}, \theta^{t+1}) - \mathcal{L}(\mathbf{V}, \theta_{-z}^t) + \mathcal{L}(\mathbf{V}, \theta^t)\Big] \\
\approx{}&\sum_{t < T} \Big\{G_\mathbf{V}^t\Big[\theta_{-z}^{t+1} - \theta_{-z}^t\Big] - G_\mathbf{V}^t\Big[\theta^{t+1} - \theta^t\Big]\Big\} \\
={}&\sum_{t < T} \Big\{G_\mathbf{V}^t\Big[\theta_{-z}^{t+1} - \theta^{t+1}\Big] - G_\mathbf{V}^t\Big[\theta_{-z}^t - \theta^t\Big]\Big\} \\
={}&\sum_{t < T} \Big\langle G_\mathbf{V}^t, \mathcal{D}^t(z)\Big\rangle,
\end{aligned}
\tag{19}
$$

where $\mathcal{D}^t(z) = \mathcal{I}_\theta^{t+1}(z) - \mathcal{I}_\theta^t(z)$ denote the sample-wise influence difference between adjacent time steps. So, we can find that the core of the estimation also lies in demonstrating the validity of Lemma 4.1 since it provides an estimation for $\mathcal{D}^t(z)$.

### F.1 PROOF FOR LEMMA 3.1

#### F.1.1 DIFF-IN FOR THE SGD OPTIMIZER

Let $\theta^t$ and $\theta_{-z}^t$ be the parameters at the $t$-th time step trained on the full-set $\mathbf{D}$ and the sub set $\mathbf{D}/z$ excluding data point $z$. So, we have that

$$
\begin{aligned}
\mathcal{D}^t(z) &= \mathcal{I}_\theta^{t+1}(z) - \mathcal{I}_\theta^t(z) \\
&= \left( \theta_{-z}^{t+1} - \theta^{t+1} \right) - \left( \theta_{-z}^t - \theta^t \right) \\
&= \left( \theta_{-z}^{t+1} - \theta_{-z}^t \right) - \left( \theta^{t+1} - \theta^t \right).
\end{aligned}
\tag{20}
$$

Considering the SGD update rules where $\eta_t$ is the learning rate and $\beta_t$ is the momentum weight, we have the following equation:

$$
\theta^{t+1} - \theta^t = -\eta_t G^t,
\tag{21}
$$

where $G^t$ is the gradient over the training set at the $t$-th step. Thus, we have the following detailed form for the influence differentiation, that is,

$$
\begin{aligned}
\mathcal{D}^t(z) &= -\eta^t \left( G_{-z}^t - G^t \right) \\
&= -\eta^t \left( G_{-z}^t - G^0 + G^0 - G^t \right) \\
&= -\eta^t \left( (G_{-z}^t - G_{-z}^0) - (G^t - G^0) - \frac{1}{N} G_z^0 \right) \\
&= -\eta^t \left( \mathcal{Q}_{-z}^t - \mathcal{Q}^t - \frac{1}{N} G_z^0 \right) \\
&\approx -\eta^t \left( \mathcal{Q}_{-z}^t - \mathcal{Q}^t \right),
\end{aligned}
\tag{22}
$$

where we introduce $\mathcal{Q}^t$ and $\mathcal{Q}_{-z}^t$ to denote the gradient difference terms $G_{-z}^k - G_{-z}^0$ and $G^k - G^0$. We disregard the term $\frac{1}{N} G_z^0$ because it is a random factor, given that the network parameter is randomly generated at the initial stage. By treating the time-step to be continuous, we can estimate the gradient difference term as:

$$
\begin{aligned}
\mathcal{Q}^t = G^t - G^0 &\approx \int_0^t \frac{\partial G^k}{\partial \theta} \frac{\partial \theta}{\partial k} d_k \\
&\approx \sum_{k=0}^t H^k \left( \theta^{k+1} - \theta^k \right) \\
&= \sum_{k=0}^t -\eta^k H^k G^k.
\end{aligned}
\tag{23}
$$

The loss function during the optimization process has the following relationship, $\frac{N}{N-1} \mathcal{L}(\mathbf{D}/z, \theta) = \mathcal{L}(\mathbf{D}, \theta) - \frac{1}{N} \ell(z, \theta)$. Since the size of the training set $N$ is generally greatly larger than 1, so $\frac{N}{N-1} \approx 1$. Hence, the above relation could be approximated as $\mathcal{L}(\mathbf{D}/z, \theta) = \mathcal{L}(\mathbf{D}, \theta) + \epsilon \ell(z, \theta)$, where $\epsilon = -\frac{1}{N}$ is a very small perturbation coefficient. Note that this approximation is also widely used in influence analysis (Koh & Liang, 2017; Cook, 1986; Yang et al., 2023; Koh et al., 2019; Grosse et al., 2023). Based on this relation, we treat the gradient difference function $\mathcal{Q}$ as a function of $\epsilon$ by following (Koh & Liang, 2017; Yang et al., 2023), that is, $\mathcal{Q}_{-z}^t = \mathcal{Q}_{\epsilon=-\frac{1}{N}}^t$ and $\mathcal{Q}^t = \mathcal{Q}_{\epsilon=0}^t$. Here, we provide the general form for $\mathcal{Q}_\epsilon^t$:

$$
\mathcal{Q}_\epsilon^t \approx \sum_{k=0}^t -\eta^k H_\epsilon^k G_\epsilon^k,
\tag{24}
$$

where $H_\epsilon^t = \nabla^2\Big[\mathcal{L}(\mathbf{D}, \theta^t) + \epsilon\ell(z, \theta^t)\Big]$ is the Hessian and $G_\epsilon^t = \nabla\Big[\mathcal{L}(\mathbf{D}, \theta^t) + \epsilon\ell(z, \theta^t)\Big]$ is the gradient. So, we can approximate $\mathcal{Q}_{-z}^t$ with $\mathcal{Q}^t$ by taking the Taylor expansion of the function with respect to the perturbation variable $\epsilon$ without retraining a model on the leave-one-out subset $\mathbf{D}/z$,

$$\mathcal{Q}_{-z}^t - \mathcal{Q}^t = \mathcal{Q}_{\epsilon=\frac{-1}{N}}^t - \mathcal{Q}_{\epsilon=0}^t \approx -\frac{1}{N}\frac{\partial\mathcal{Q}_\epsilon^t}{\partial\epsilon}\Big|_{\epsilon=0}. \tag{25}$$

Obviously, $\frac{\partial\mathcal{Q}_\epsilon^t}{\partial\epsilon}\Big|_{\epsilon=0}$, the derivative of $\mathcal{Q}_\epsilon^t$ with respect to the perturbation variable $\epsilon$ can be easily obtained, that is,

$$\frac{\partial\mathcal{Q}_\epsilon^t}{\partial\epsilon}\Big|_{\epsilon=0} = -\sum_{k=0}^t \eta^k H^k \nabla\ell(z, \theta^k) - \sum_{k=0}^t \eta^k \nabla^2\ell(z, \theta^k)G^k, \tag{26}$$

where we use simple expression $H^t$ to denote the Hessian $H_{\mathbf{D}}^t$ and use the short expression $G^t$ to denote the gradient $G_{\mathbf{D}}^t$. By substituting this formula into Eq.32, we obtain the estimator for the influence difference term $\mathcal{D}^t(z)$, that is,

$$\begin{aligned}
\mathcal{D}^t(z) \\
&\approx -\eta^t\Big(\mathcal{Q}_{-z}^t - \mathcal{Q}^t\Big) \\
&\approx \sum_{k=0}^t \frac{-\eta^t\eta^k}{N}\Big(H^k G_z^k + H_z^k G^k\Big),
\end{aligned} \tag{27}$$

where $G_z^k = \nabla\ell(z, \theta^k)$ is the gradient over sample $z$ at $k$-th step and $H_z^k = \nabla^2\ell(z, \theta^k)$ is the gradient over sample $z$ at $k$-th step.

### F.1.2 DIFF-IN FOR THE MOMENTUM-BASED OPTIMIZER

Let $\theta^t$ and $\theta_{-z}^t$ be the parameters at the $t$-th time step trained on the full-set $\mathbf{D}$ and the sub set $\mathbf{D}/z$ excluding data point $z$. So, we have that

$$\begin{aligned}
\mathcal{D}^t(z) &= \mathcal{I}_\theta^{t+1}(z) - \mathcal{I}_\theta^t(z) \\
&= \Big(\theta_{-z}^{t+1} - \theta^{t+1}\Big) - \Big(\theta_{-z}^t - \theta^t\Big) \\
&= \Big(\theta_{-z}^{t+1} - \theta_{-z}^t\Big) - \Big(\theta^{t+1} - \theta^t\Big).
\end{aligned} \tag{28}$$

Considering the SGD-M update rules where $\eta_t$ is the learning rate and $\beta_t$ is the momentum weight, we have the following equation:

$$\begin{aligned}
\theta^{t+1} - \theta^t &= -\eta_t\Big((1-\beta_t)G^t + \beta_t(\theta^t - \theta^{t-1})\Big) \\
&= \sum_{k=0}^t \Big(\prod_{k<a<t}\eta_a\beta_a\Big)\eta_k(\beta_k - 1)G^k,
\end{aligned} \tag{29}$$

where $G^t$ is the gradient over the training set at the $t$-th step. By using $\alpha_k^t$ to denote $\Big(\prod_{k<a<t}\eta_a\beta_a\Big)\eta_k(\beta_k - 1)$, we obtain the short formulation for the rule:

$$\theta^{t+1} - \theta^t = \sum_{k=0}^t \alpha_k^t G^k, \tag{30}$$

Thus, we have the following detailed form for the influence differentiation, that is,

$$\mathcal{D}^t(z) = \sum_{k=0}^t \alpha_k^t\Big(G_{-z}^k - G^k\Big). \tag{31}$$

To obtain the precise calculation of the gradient $G_{-z}^k$, it is necessary to perform the extremely costly LOO retraining. Therefore, we must rely on estimation methods. Here, we introduce an intermediate

term $G^0 = G^0_{-z} + \frac{1}{N}G^0_z$, where $G^0_z$ is the gradient over the data point $z$ at the initial step. Hence, the influence differentiation could be rewritten as:

$$
\begin{aligned}
\mathcal{D}^t(z) &= \sum_{k=0}^{t} \alpha_k^t \left( G^k_{-z} - G^0 + G^0 - G^k \right) \\
&= \sum_{k=0}^{t} \alpha_k^t \left( (G^k_{-z} - G^0_{-z}) - (G^k - G^0) - \frac{1}{N}G^0_z \right) \\
&= \sum_{k=0}^{t} \alpha_k^t \left( \mathcal{Q}^k_{-z} - \mathcal{Q}^k - \frac{1}{N}G^0_z \right) \\
&\approx \sum_{k=0}^{t} \alpha_k^t \left( \mathcal{Q}^k_{-z} - \mathcal{Q}^k \right),
\end{aligned}
\tag{32}
$$

where we introduce $\mathcal{Q}^k$ and $\mathcal{Q}^k_{-z}$ to denote the gradient difference terms $G^k_{-z} - G^0_{-z}$ and $G^k - G^0$. We disregard the term $\frac{1}{N}G^0_z$ because it is essentially a random factor, given that the network parameter is randomly generated at the initial stage.

By treating the time-step to be continuous, we can estimate the gradient difference term as:

$$
\begin{aligned}
\mathcal{Q}^t = G^t - G^0 &\approx \int_0^t \frac{\partial G^k}{\partial \theta} \frac{\partial \theta}{\partial k} d_k \\
&\approx \sum_{k=0}^{t} H^k \left( \theta^{k+1} - \theta^k \right).
\end{aligned}
\tag{33}
$$

By substituting the concrete form of $\theta^{t+1} - \theta^t$ in Eq.30 into gradient difference, we have

$$
\mathcal{Q}^t \approx \sum_{k=0}^{t} H^k \sum_{q=0}^{k} \alpha_q^k G^q,
\tag{34}
$$

where $\alpha_q^k = \left( \prod_{q<a<k} \eta_a \beta_a \right) \eta_q(\beta_q - 1)$ is the coefficient. The loss function during the optimization process has the following relationship, $\frac{N}{N-1}\mathcal{L}(\mathbf{D}/z, \theta) = \mathcal{L}(\mathbf{D}, \theta) - \frac{1}{N}\ell(z, \theta)$. Since the size of the training set $N$ is generally greatly larger than 1, so $\frac{N}{N-1} \approx 1$. Hence, the above relation could be approximated as $\mathcal{L}(\mathbf{D}/z, \theta) = \mathcal{L}(\mathbf{D}, \theta) + \epsilon\ell(z, \theta)$, where $\epsilon = -\frac{1}{N}$ is a very small perturbation coefficient. Note that this approximation is also widely used in influence analysis (Koh & Liang, 2017; Cook, 1986; Yang et al., 2023; Koh et al., 2019; Grosse et al., 2023). Based on this relation, we treat the gradient difference function $\mathcal{Q}$ as a function of $\epsilon$ by following (Koh & Liang, 2017; Yang et al., 2023), that is, $\mathcal{Q}^t_{-z} = \mathcal{Q}^t_{\epsilon=\frac{-1}{N}}$ and $\mathcal{Q}^t = \mathcal{Q}^t_{\epsilon=0}$. Here, we provide the form for $\mathcal{Q}^t_{\epsilon}$:

$$
\mathcal{Q}^t_{\epsilon} \approx \sum_{k=0}^{t} H^k_{\epsilon} \sum_{q=0}^{k} \alpha_q^k G^q_{\epsilon},
\tag{35}
$$

where $H^t_{\epsilon} = \nabla^2 \left[ \mathcal{L}(\mathbf{D}, \theta^t) + \epsilon\ell(z, \theta^t) \right]$ is the Hessian and $G^t_{\epsilon} = \nabla \left[ \mathcal{L}(\mathbf{D}, \theta^t) + \epsilon\ell(z, \theta^t) \right]$ is the gradient.

So, we can approximate $\mathcal{Q}^t_{-z}$ with $\mathcal{Q}^t$ by taking the Taylor expansion of the function with respect to the perturbation variable $\epsilon$ without retraining a model on the leave-one-out subset $\mathbf{D}/z$,

$$
\mathcal{Q}^t_{-z} - \mathcal{Q}^t = \mathcal{Q}^t_{\epsilon=\frac{-1}{N}} - \mathcal{Q}^t_{\epsilon=0} \approx -\frac{1}{N} \frac{\partial \mathcal{Q}^t_{\epsilon}}{\partial \epsilon}\bigg|_{\epsilon=0}.
\tag{36}
$$

Obviously, $\frac{\partial \mathcal{Q}^t_{\epsilon}}{\partial \epsilon}\big|_{\epsilon=0}$, the derivative of $\mathcal{Q}^t_{\epsilon}$ with respect to the perturbation variable $\epsilon$ can be easily obtained, that is,

$$
\begin{aligned}
&\frac{\partial \mathcal{Q}^t_{\epsilon}}{\partial \epsilon}\bigg|_{\epsilon=0} \\
&= \sum_{k=0}^{t} H^k \sum_{q=0}^{k} \alpha_q^k \nabla\ell(z, \theta^q) + \sum_{k=0}^{t} \nabla^2\ell(z, \theta^k) \sum_{q=0}^{k} \alpha_q^k G^q,
\end{aligned}
\tag{37}
$$

where we use simple expression $H^t$ to denote the Hessian $H^t_{\mathbf{D}}$ and use the short expression $G^t$ to denote the gradient $G^t_{\mathbf{D}}$. By substituting this formula into Eq.32, we obtain the estimator for the influence difference term $\mathcal{D}^t(z)$, that is,

$$
\begin{aligned}
\mathcal{D}^t(z) \\
\approx \sum_{k=0}^{t} \alpha_k^t \left( \mathcal{Q}_{-z}^k - \mathcal{Q}^k \right) \\
\approx \sum_{k=0}^{t} \frac{-\alpha_k^t}{N} \left( \sum_{q=0}^{k} H^q \sum_{e=0}^{q} \alpha_e^q \nabla \ell(z, \theta^e) + \sum_{q=0}^{k} \nabla^2 \ell(z, \theta^q) \sum_{e=0}^{q} \alpha_e^q G^e \right) \\
\approx \sum_{k=0}^{t} \frac{-\alpha_k^t}{N} \left( \sum_{q=0}^{k} H^q \sum_{e=0}^{q} \alpha_e^q G_z^e + \sum_{q=0}^{k} H_z^q \sum_{e=0}^{q} \alpha_e^q G^e \right),
\end{aligned}
\tag{38}
$$

where $G_z^q = \nabla \ell(z, \theta^q)$ is the gradient over sample $z$ at $q$-th step and $H_z^k = \nabla^2 \ell(z, \theta^k)$ is the gradient over sample $z$ at $k$-th step. And $\alpha_k^t = \left( \prod_{k<a<t} \eta_a \beta_a \right) \eta_k (\beta_k - 1)$ is a coefficient defined by the learning rate $\eta$ at each step and the momentum weight $\beta$ at each step.

If we update the definition of $\alpha_k^t$ by set to $\alpha_k^t \leftarrow \frac{-1}{N} \alpha_k^t = \frac{1}{N} \left( \prod_{k<a<t} \eta_a \beta_a \right) \eta_k (1 - \beta_k)$, then, we have,

$$
\mathcal{D}^t(z) \approx \sum_{k=0}^{t} \alpha_k^t \left( \sum_{q=0}^{k} H^q \sum_{e=0}^{q} \alpha_e^q G_z^e + \sum_{q=0}^{k} H_z^q \sum_{e=0}^{q} \alpha_e^q G^e \right),
\tag{39}
$$

## G    Proof for Proposition 4.1

As shown in the last section, the core for the calculation is using Lemma 4.1 to estimate the influence difference term denoted by $\mathcal{D}^t(z)$, where we use Eq.32 to estimate the gradient difference term $\mathcal{Q}^k = G^k - G^0$. This term is also the primary source of approximation error. Let $\mathcal{D}^{t*}(z)$ and $\mathcal{D}^t(z)$ to denote the actual term and our estimated term, respectively. Therefore, we can formulate the error term as follows:

$$
\begin{aligned}
\left| \mathcal{D}^{t*}(z) - \hat{\mathcal{D}}^t(z) \right| \\
= \left| \sum_{k=0}^{t} \alpha_k^t \left( \mathcal{Q}_{-z}^{k*} - \mathcal{Q}^{k*} - \frac{1}{N} G_z^0 - \left( \mathcal{Q}_{-z}^k - \mathcal{Q}^k \right) \right) \right| \\
\leqslant 2T \left| \mathcal{Q}^{k*} - \mathcal{Q}^k \right| + \frac{T}{N} \left| G_z^0 \right|,
\end{aligned}
\tag{40}
$$

which always holds by assuming $\alpha_k^t \leqslant 1$. And this assumption is generally true because $\alpha_k^t$ is obtained by multiplying the historical learning rate $\eta$ and the weight of the momentum item $\beta$, where $\eta$ and $\beta$ are both real-valued float numbers generally within $(0, 1)$.

By assuming the gradient is upper-bounded by $g$, so, we obtain

$$
\left| \mathcal{D}^{t*}(z) - \mathcal{D}^t(z) \right| \leqslant 2T \left| \mathcal{Q}^{k*} - \hat{\mathcal{D}}^k \right| + \frac{T}{N} g,
\tag{41}
$$

where $\mathcal{Q}^{k*} = G^k - G^0$. Compared with the actual $\mathcal{Q}^{k*}$, the error of $\mathcal{Q}^k$ comes from the Eq.33, where we use $G^k$ to estimate $G^0$ to establish the functional relation between $\mathcal{Q}^k$ and the perturbation variable $\epsilon$ (see Eq.35 for more details). Hence, by using $\hat{G}^0$ to denote the estimated one, we can further mine the error inequality:

$$
\begin{aligned}
\left| \mathcal{D}^{t*}(z) - \hat{\mathcal{D}}^t(z) \right| &\leqslant 2T \left| \mathcal{Q}^{k*} - \mathcal{Q}^k \right| + \frac{T}{N} g \\
&= 2T \left| G^0 - \hat{G}^0 \right| + \frac{T}{N} g \\
&= 2T \left| G^0 - G^k + \sum_{k=0}^{t} H_\epsilon^k \sum_{q=0}^{k} \alpha_q^k G^q \right| + \frac{T}{N} g \\
&\leqslant 2T \left| G^0 - G^k \right| + 2T \left| \sum_{k=0}^{t} H_\epsilon^k \sum_{q=0}^{k} \alpha_q^k G^q \right| + \frac{T}{N} g.
\end{aligned}
\tag{42}
$$

Assuming the gradient is Lipschitz-continuous, we have

$$
\begin{aligned}
\left|\mathcal{D}^{t*}(z) - \hat{\mathcal{D}}^t(z)\right| &\leqslant 2T\left|G^0 - G^k\right| + 2T\left|\sum_{k=0}^{t} H_\epsilon^k \sum_{q=0}^{k} \alpha_q^k G^q\right| + \frac{T}{N}g \\
&\leqslant 2T\left|G^0 - G^k\right| + 2T\sum_{k=0}^{t}\left|H_\epsilon^k \sum_{q=0}^{k} \alpha_q^k G^q\right| + \frac{T}{N}g \\
&\leqslant 2T\left|G^0 - G^k\right| + 2T\ell\sum_{k=0}^{t}\left|\sum_{q=0}^{k} \alpha_q^k G^q\right| + \frac{T}{N}g \\
&\leqslant 2T\left|G^0 - G^k\right| + 2T^2\ell \max_k\left|\sum_{q=0}^{k} \alpha_q^k G^q\right| + \frac{T}{N}g \\
&\leqslant 2T\ell \max_{t\leqslant T}||\theta^t - \theta^0|| + 2T^2\ell \max_k\left|\sum_{q=0}^{k} \alpha_q^k G^q\right| + \frac{T}{N}g.
\end{aligned}
\tag{43}
$$

By using the equation of the difference of the parameters between time steps in Eq.30, that is, $\theta^{t+1} - \theta^t = \sum_{k=0}^{t} \alpha_k^t G^k$. By assuming $\max_{t\leqslant T}||\theta^t - \theta^0|| \gg \max_{t<T}||\theta^{t+1} - \theta^t||$, we can further derive this error bound,

$$
\begin{aligned}
\left|\mathcal{D}^{t*}(z) - \hat{\mathcal{D}}^t(z)\right| &\leqslant 2T\ell \max_{t\leqslant T}||\theta^t - \theta^0|| + 2T^2\ell \max_{t<T}||\theta^{t+1} - \theta^t|| + \frac{T}{N}g \\
&\leqslant 2T\ell(T+1)C + \frac{T}{N}g,
\end{aligned}
\tag{44}
$$

where we use the notation $C$ to indicate the maximum parameter gap $\max_{t\leqslant T}||\theta^t - \theta^0||$. Finally, we have

$$
\begin{aligned}
|\mathcal{I}(z) - \mathcal{I}^{\text{Exact}}(z)| &\leqslant |\sum_t \mathcal{D}^t * (z) - \hat{\mathcal{D}}^t(z)| \\
&\leqslant T\max_t\left|\mathcal{D}^{t*}(z) - \hat{\mathcal{D}}^t(z)\right| \\
&\leqslant 2T^2\ell(T+1)C + \frac{T^2}{N}g.
\end{aligned}
\tag{45}
$$

