# OpenReview forum: "Diff-In: Data Influence Estimation with Differential Approximation"
_ICLR.cc/2025/Conference — Submitted to ICLR 2025_

### Official Review · Reviewer_3Kxb · 2024-11-01

**Soundness:** 4
**Presentation:** 4
**Contribution:** 4
**Rating:** 6
**Confidence:** 4

**Summary:**

This paper introduces a novel method called Diff-In for estimating the influence of data points during model training. Unlike existing methods that rely on convexity assumptions or first-order approximations, Diff-In calculates the cumulative differences in influence between consecutive training steps. By using second-order approximations of the Hessian-gradient product, it achieves higher accuracy without increasing computational complexity. The approach is scalable and can be applied to large datasets. Extensive experiments demonstrate that Diff-In outperforms previous methods in data-centric tasks like data cleaning, data deletion, and coreset selection, particularly excelling in large-scale vision-language pretraining tasks.

**Strengths:**

- Innovative Approach: Diff-In offers a fresh perspective by focusing on the temporal differences in influence, which significantly improves accuracy without relying on model convexity. Moreover, Diff-In achieves second-order estimation accuracy with computational complexity similar to first-order methods, combining high accuracy with efficiency.
- Scalability: The method is computationally efficient and scalable to large datasets, making it suitable for modern machine learning applications.
- Comprehensive Evaluation: The paper presents extensive experiments on various tasks, demonstrating Diff-In’s superior performance over existing methods across several benchmarks.

**Weaknesses:**

- Polynomial growth of estimation error: The theoretical upper limit of the Diff-In estimation error grows polynomially with the number of training steps. This growth may still limit the scalability of the method in longer training scenarios. Exploring the actual trade-off between training steps T and error can enhance the applicability of the method.
- Matching between data quality and model capabilities: Using influence scores directly for data management does not seem to take into account the matching of data quality and model capabilities. For example, when performing core set selection, can only considering the influence score alone really select a high-quality data subset or a subset that is most helpful for downstream tasks?

**Questions:**

None

---

> ### Author Response · Authors · 2024-11-23
> **Response to Reviewer 3Kxb**
>
> Thank you for your time and effort during the review process, as well as for your recognition and thoughtful suggestions regarding our paper. Your feedback is highly valuable and has significantly contributed to improving the quality of our paper.
>
> ---
>
> **Weakness-1. Concerns about the polynomial growth of estimation error.**
>
> Thanks for your constructive comments!
>
> (a) This phenomenon is intuitive. The reason that increasing the training steps leads to larger theoretical errors is that the optimized parameters diverge further from the initialized parameters as training progresses. This divergence makes accurate estimation more challenging and contributes to the accumulation of approximation errors.
>
> It is also worth noting that some methods, such as those in [1, 2], exhibit a faster increase in error as $T$ grows. In contrast, our approach, with its polynomial error bound, demonstrates a significantly smaller growth rate compared to other methods, such as the exponential growth observed with SGD-Inf [2]. This indicates that our method effectively mitigates error accumulation over time, even as $T$ increases.
>
>
> (b) It is important to emphasize that this bound reflects a worst-case scenario for error. In practice, Diff-In performs robustly even with larger $T$ values. As training progresses and the model approaches convergence, parameter changes become minimal, meaning larger $T$ typically has a negligible impact on errors. This highlights the effectiveness of our approach in maintaining stable performance.
>
> (c) We present results comparing performance across different values of $T$. The table below illustrates how Diff-In performs relative to the maximum training time $T$ in coreset selection experiments on ImageNet. Notably, increasing $T$ does not degrade Diff-In's practical performance.
>
>
> Method|T=50|T=100|T=200
> ---|---|---|---
> **Performance of coreset by Diff-In (SR=20\%)**|61.7|61.4|62.4|
>
> [1]. Theoretical and practical perspectives on what influence functions do. NeurIPS 2024.
>
> [2]. Data cleansing for models trained with sgd. NeurIPS 2019
>
>
>
>
>
> **W-2. Concerns about the generalization ability of the samples selected by Diff-In.**
>
> Thank you for your helpful comments!
>
> We believe that the coreset selected by Diff-In can be used for different capabilities of models and its generalization on downstream tasks.
>
> As demonstrated in Section 5.3, "Coreset Selection for Vision-Language Pretraining," the subsets chosen by Diff-In enable pre-trained models to achieve superior performance across various tasks, including zero-shot classification, linear probing classification, and image/text retrieval. This improvement stems from the fact that the utility of data often depends on the inherent quality of the samples, which is relatively independent of the model's capabilities.
>
> ---
>
> **We would be happy to discuss this further if you have any concerns.**

---

> > ### Comment · Reviewer_3Kxb · 2024-11-24
> >
> > Thank you for your response and additional experimental results. Your reply has clarified some of my concerns. After careful consideration, I will keep the score as it is.
> >
> > Best regards,
> > Reviewer 3Kxb

---

> > > ### Author Response · Authors · 2024-11-25
> > > **A grateful response to Reviewer 3Kxb**
> > >
> > > Thank you for your response!
> > >
> > > We appreciate your recognition and valuable feedback. If any aspects of the paper need clarification or if you have suggestions for improvement, we would be eager to address them. Your insights are important to us, and we are committed to making any necessary adjustments to enhance our submission. Thank you once again!
> > >
> > > Best regards,
> > >
> > > Authors of Submission 1257

---

### Official Review · Reviewer_gKRM · 2024-11-02

**Soundness:** 3
**Presentation:** 3
**Contribution:** 3
**Rating:** 6
**Confidence:** 3

**Summary:**

This paper introduces Diff-In, a novel approach for estimating the influence of training samples across multiple learning steps. The core innovation of Diff-In lies in approximating the second-order Hessian-gradient product using only first-order computations. This approach enables Diff-In to achieve the high accuracy typical of second-order methods while maintaining the computational efficiency of first-order techniques.  Theoretical analysis and extensive experiments support Diff-In's effectiveness.

**Strengths:**

- The paper is well-written and clearly structured, making the methodology easy to follow.
- The formulation is intuitive, and the theoretical analysis appears generally sound.
- Experimental results show Diff-In's effectiveness.

**Weaknesses:**

- Diff-In closely follows TraceIn’s approach of accumulating influence over successive training steps. The main technical advancement appears to be an extension of TraceIn’s formulation (Eq. 3) to Eq. 5, limiting its novelty.
- The baseline choices are somewhat outdated, and the authors should include comparisons with more recent state-of-the-art methods, such as [1-2].
- Additionally, speed comparisons with these newer approaches are missing, which would provide a more comprehensive evaluation.

[1] Studying Large Language Model Generalization with Influence Functions

[2] Trak: Attributing Model Behavior at Scale

**Questions:**

- The authors evaluate Diff-In through downstream applications but do not consider counterfactual-type metrics, such as the Linear Data Modeling Score (LDS) introduced in [2]. Would such metrics provide further insight into Diff-In's performance?
- While Figure 3 demonstrates that Diff-In incurs minimal computational overhead compared to TracIn, this may be limited to smaller models. Could the authors provide similar speed comparisons for large-scale models?

---

> ### Author Response · Authors · 2024-11-23
> **Response to Reviewer gKRM (Weakness part)**
>
> Thank you for your time and effort in reviewing our paper. We sincerely appreciate your recognition and thoughtful suggestions. Your feedback is valuable and has greatly enhanced the quality of our work.
>
> ---
>
> **Response to Weaknesses**
>
> **W-1. Diff-In closely follows TraceIn’s approach of accumulating influence over successive training steps.**
>
> Thank you for bringing this up! While both TracIn and Diff-In accumulate influence over successive training steps, they differ significantly in their objectives, formulations, and outcomes, as summarized below.
>
> (a) Objectives: Diff-In is designed to estimate the impact of removing a sample on the model's performance, known as leave-one-out influence. This measure can be used to assess a data sample's influence on both the loss and model parameters (see Sec.3.2). In contrast, TracIn focuses on estimating changes in the validation loss when a sample is excluded from training during specific iterations (referred to as TracInIdeal, see Sec. 2.1). However, TracIn does not guarantee an accurate estimation of leave-one-out influence.
>
>
> (b) Formulation: Their distinct objectives lead to differences in their mathematical formulations. Diff-In defines sample-wise influence on model parameters as the **cumulative sum of the parameter changes** across successive training iterations. This difference term is approximated using **second-order derivative information** with high accuracy, eliminating the need for the model convexity assumptions required by existing methods.
> In contrast, TracIn estimates a sample's influence on loss values for a given dataset by **summing its impact on the loss** across various training iterations. This estimation relies on first-order gradient information for approximation.
> These differences become particularly evident when advanced optimizers, such as Adam or SGD with momentum, are used, as elaborated in Appendix B.
>
>
> (c) Performance:  These differences result in distinct outcomes for TracIn and Diff-In. TracIn is limited to estimating a sample's influence on loss values, whereas Diff-In can estimate the influence on both parameters and loss. Consequently, Diff-In supports a broader range of tasks, such as coreset, data cleaning, and data deletion/unlearning, while TracIn is restricted to coreset and data cleaning tasks. Furthermore, Diff-In consistently outperforms TracIn on data cleaning on GSM8K and coreset selection across ImageNet-1K, CC12M.
>
>
>
> **W-2 and W-3. Performance and speed comparison with some new works on large-scale datasets and large models.**
>
>
> Thank you for the constructive comment!
>
> During the rebuttal period, we conducted coreset experiments to compare Diff-In with EK-FAC [1] and TRAK [2]. These experiments utilized the Llama-3-8B-SFT model with a selection rate of 10\% on the OpenMathInstruct dataset, which contains 1.8 million math problems and solutions. Additional details are provided in Appendix D.4.
>
> Diff-In outperformed TracIn and TRAK by 3.1\% and 1.7\%, respectively. Although EK-FAC proved to be the most computationally efficient method, its performance was significantly lower due to its reliance on the convexity assumption, which is not well-suited for deep models. Overall, the computation time across all methods was comparable, efficient, and scalable to large-scale models.
>
>
> Method|8-shot (CoT) on GSM8K|Selection time cost
> ---|---|---
> Llama-3-8B-Instruct|77.2|-
> SFT on 1.8M data|85.3|-
> SR=30\% by EK-FAC[1]|80.3|18.4 GPU-hours
> SR=30\% by TRAK[2]|82.7|21.7 GPU-hours
> SR=30\% by TracIn|81.3|26.3 GPU-hours
> SR=30\% by Diff-In|84.4|28.7 GPU-hours
>
> [1] Studying Large Language Model Generalization with Influence Functions. ArXiv 2308.03296.
>
> [2] Trak: Attributing Model Behavior at Scale. ICML 2023.

---

> ### Author Response · Authors · 2024-11-23
> **Response to Reviewer gKRM (Question part)**
>
> **Response to Questions**
>
> **Q-1. More results with the Linear Data Modeling Score (LDS) metric.**
>
> Thank you for the question! LDS is a valuable metric for evaluating the effectiveness of data attribution methods by measuring how accurately they can predict outcomes. Here, we also use LDS to assess Diff-In, TracIn, Arnoldi-IF [1], TRAK [2], and IF on the ImageNet-1K dataset. The results demonstrate that Diff-In and TRAK consistently outperform the other methods.
>
> Method|LDS
> ---|---
> IF|0.003
> Arnoldi-IF|0.045
> TracIn|0.179
> Trak|0.238
> Diff-In|0.265
>
>
> [1]. Scaling Up Influence Functions. AAAI 2022.
>
> [2]. TRAK: Attributing Model Behavior at Scale. ICML 2023.
>
> [3]. Estimating Training Data Influence by Tracing Gradient Descent. NeurIPS 2020.
>
>
> **Q-2. Could the authors provide similar speed comparisons for large-scale models?**
>
> Thanks! Please refer to the response to W-2 and W-3.

---

> > ### Comment · Reviewer_gKRM · 2024-11-24
> >
> > Thank you for your response. I have carefully reviewed the authors' reply and appreciate the additional details provided. I encourage the authors to include the results of large-scale models/datasets in the final version of the paper to enhance its impact. After considering the discussion, I will maintain my current score.

---

> ### Author Response · Authors · 2024-11-25
> **A Grateful Response to Reviewer gKRM**
>
> Thank you for your thoughtful review and response. We appreciate your feedback and recognize the value of your insights in enhancing our work.
>
> **If there are any aspects of the paper that require clarification or if you have suggestions for improvement, we would be eager to address them. Your input is valuable to us, and we are committed to making any necessary adjustments to strengthen our submission.**
>
> Additionally, if you feel that our revisions adequately address your concerns, we would be grateful if you could consider raising the score.
>
> Thank you once again for your time and consideration!
>
> ---
>
> **We have already included those new experimental results in the paper along with those discussions of general interest to the article during the rebuttal phase. Specifically, we have made the following changes:**
>
>     1. Fixed typos (main paper).
>
>     2. Added new references [1-13] in the related work section (appendix, Sec.C).
>
>     3. Added clarification of Data deletion and new baseline [14, 15] (main paper).
>
>     4. Added speed and performance comparison with SOTA methods on experiments for Llama-3-8B on OpenMathInstruct (appendix, Sec.D.4.1 and Table 11).
>
>     5. Added further ablation study on the choice of the number of selected checkpoints (appendix, Sec.D.4.2 and Table.11 and Table 13).
>
>     6. Added further performance comparison with IP and OGI on the cleaning experiments for GSM8K  (appendix, Sec.D.4.1 and Table.12).
>
>     7. Added guidelines for checkpoint selection (appendix, Sec.D.4.3).
>
>     8. Added experiments and discussion about the growth of the bound (appendix, Sec.D.5.1).
>
>     9. Added discussion about the assumption of the bound (appendix, Sec.D.5.2).
>
>     10. Moved the discussion of limitations to the appendix.
>
> **Reference:**
>
>     [1]. Studying Large Language Model Generalization with Influence Functions. Arxiv 2308.03296.
>
>     [2]. TRAK: Attributing Model Behavior at Scale. ICML 2023.
>
>     [3]. Scaling Up Influence Functions. AAAI 2022.
>
>     [4]. Revisit, Extend, and Enhance Hessian-Free Influence Functions. Arxiv 2405.17490
>
>     [5]. Outlier Gradient Analysis: Efficiently Improving Deep Learning Model Performance via Hessian-Free Influence Functions. Arxiv 2405.03869.
>
>     [6]. DataInf: Efficiently Estimating Data Influence in LoRA-tuned LLMs and Diffusion Models. ICLR 2024.
>
>     [7]. A Survey of Machine Unlearning. Arxiv 2209.02299.
>
>     [8]. Datamodels: Predicting Predictions from Training Data. ICML 2022.
>
>     [9]. What is Your Data Worth to GPT? LLM-Scale Data Valuation with Influence Functions. ArXiv 2405.13954
>
>     [10]. "What Data Benefits My Classifier?" Enhancing Model Performance and Interpretability through Influence-Based Data Selection. ICLR 2024.
>
>     [11]. HYDRA: Hypergradient Data Relevance Analysis for Interpreting Deep Neural Networks. AAAI 2021.
>
>     [12]. Achieving Fairness at No Utility Cost via Data Reweighing with Influence. ICML 2022.
>
>     [13]. Revisiting inverse Hessian vector products for calculating influence functions. Arxiv 2409.17357
>
>     [14]. Efficiently Erasing Samples from Trained Machine Learning Models. NeurIPS 2021 Workshop.
>
>     [15]. Machine Unlearning Method Based On Projection Residual. DSSA 2022.
>
> Thank you very much again!

---

### Official Review · Reviewer_7tcF · 2024-11-03

**Soundness:** 3
**Presentation:** 3
**Contribution:** 3
**Rating:** 6
**Confidence:** 4

**Summary:**

The paper proposes a new formulation for influence estimation by accumulating differences in influence computation over iterative training timesteps, using second order approximations for the difference terms. The authors also conduct experiments in 3 task settings (data cleaning, data deletion, and coreset selection) and report performance alongside other influence computation methods and baselines.

**Strengths:**

- The approach to estimate influence by a second order approximation of the finite difference terms is novel.
- The results showcase that Diff-In can improve performance compared to the other methods considered in the paper.

**Weaknesses:**

Overall, I like the direction of the work, but find that it possesses certain issues, such as the (random) selection of checkpoints, missing experimental analysis and comparisons with other relevant influence methods, that would allow me to recommend acceptance. I am happy to engage more on the points listed below:

1. **Selecting Checkpoints (i.e. choosing $m$)**: I find the (hyperparameter) step of selecting checkpoints to be one of the major drawbacks with the approach as this is implicitly assumed by the method. While the authors conduct experiments on CIFAR-10 and ResNet-18 in Section 5.4 (and in Figure 2 on sampling strategies), these recommendations cannot generalize to new datasets and ideally this step will need to be carried out prior to utilizing Diff-In. That is, the appropriate value of $m$ might be different for other datasets and models, and it essentially becomes a hyperparameter in need of optimization. This step can be prohibitively expensive to undertake for models with a large number of parameters (e.g. LLMs). Furthermore, the authors recommend using random sampling to select the relevant set of checkpoints $\mathcal{T}_m$ which introduces another potential issue. It is possible, especially with small $m$ (i.e. $m=5$) that all the checkpoints chosen can lead to inaccurate estimation. That is, for the same setting of $m=5$, a practitioner utilizing Diff-In could obtain very different results, and it is not clear to me how this randomness in performance can be mitigated. Ideally, ablations on other datasets and models need to be undertaken to showcase if indeed the value of $m$ is general across datasets/models, which seems unlikely.
2. **Missing Experimental Analysis**: First, I find that the paper is lacking more extensive comparison with other relevant influence based approaches. To truly assess the benefits of Diff-In, the authors should compare with recent influence approaches designed for larger models (such as EK-FAC [1], TRAK [2], and/or Arnoldi iteration [3]) and better Hessian-free methods (for instance, IP [4] and outlier gradient influence [5] seem to be very relevant for the data cleaning task). I think it is especially important to consider methods that circumvent the convexity assumption as we are dealing with larger models-- for e.g. EK-FAC estimates PBRF instead of LOO by using the Gauss-Newton Hessian instead of the standard Hessian. Second, I believe that for the data cleaning experiments of Section 5.1 the authors should consider standard influence estimation alongside self-influence as in prior work [5,6]. What are the results when the influence is computed on a clean validation set? Is Diff-In still the top performer? Third, since the paper claims that Diff-In remains scalable across larger models, I believe it is necessary to have more comprehensive analysis of running time especially with larger models and other methods. Specifically, running time analysis should be conducted on other datasets/models (preferably larger models) where both performance and running time should be provided for comparison. I also feel that the time taken to select the ideal number of checkpoints $m$ (if not fixed as 5) should be factored in as well.
3. **Data Deletion and Machine Unlearning**: The data deletion setting seems to me to be conceptually equivalent to undertaking machine unlearning [7] and as the unlearning community has studied this problem extensively, it would be useful to utilize some recent and popular approaches (refer to [7]) for fair comparison (and not just 2 influence approaches). I would also suggest that the authors reframe this subsection and task to reflect that it is conceptually just unlearning (or otherwise list out the key differences). Data deletion in past work [5,6] has usually allowed one model retraining, and so that readers are not confused I think this should be made clear both in name and through better descriptions by referring to it as unlearning.
4. **Adding to Related Works**: Despite a comprehensive related works in the appendix, I think there are a number of papers that should also be discussed that are currently missing. Alongside a few of the papers mentioned above (for experimental comparison), it would also be beneficial to include descriptions on methods such as Datamodels [8], LoGra [9], tree estimation [10], HyDRA [11], data reweighing [12], among other relevant work in this space [13,14].
5. **Typos**: There are some typos in the paper that can be revised. For instance on line 84 (page 2), "TraceIn" -> "TracIn", and the section heading for Section 3 should read "Appromati" -> "Approximation". There are also repeated references: for instance, the paper "On second-order group influence functions for black-box predictions" by Basu et al appears twice in the bibliography.

**References**:
1. https://arxiv.org/pdf/2308.03296
2. https://arxiv.org/pdf/2303.14186
3. https://arxiv.org/pdf/2112.03052
4. https://arxiv.org/pdf/2405.17490
5. https://arxiv.org/pdf/2405.03869
6. https://arxiv.org/pdf/2310.00902
7. https://arxiv.org/pdf/2209.02299v6
8. https://arxiv.org/pdf/2202.00622
9. https://arxiv.org/pdf/2405.13954
10. https://openreview.net/pdf?id=HE9eUQlAvo
11. https://arxiv.org/pdf/2102.02515
12. https://proceedings.mlr.press/v162/li22p/li22p.pdf
13. https://arxiv.org/pdf/2409.17357
14. https://openaccess.thecvf.com/content/ICCV2021/papers/Liu_Influence_Selection_for_Active_Learning_ICCV_2021_paper.pdf

**Questions:**

Each of the weaknesses listed above can be considered as a question.

---

> ### Author Response · Authors · 2024-11-23
> **Response to Reviewer 7tcF (Part-1)**
>
> Thank you for your time and effort during the review process, as well as for your recognition and thoughtful suggestions regarding our paper. Your feedback is highly valuable and has significantly contributed to improving the quality of our paper.
>
> ---
>
> **Weakness.1-1. Questions on the hyperparameter (the number of selected time steps and the sampling method):  generalization across different datasets.**
>
> (a). The number of checkpoints and the sampling method was optimized using the CIFAR-10 dataset and a ResNet-18 backbone, with results presented in Figures 3(a) and 3(b) in Sec. 5.4. These results demonstrate that setting the number of checkpoints $m = 5$ provides the best trade-off between accuracy and efficiency, while uniform sampling across all learning stages achieves the optimal performance. Consequently, we adopt $m = 5$ and uniform sampling for checkpoint selection in all our experiments.
>
> Our model's strong performance across three tasks (e.g., coreset selection, data cleaning, data deletion) and eight datasets (e.g., ImageNet-1K/22K, CC12M, GSM8K, OpenMathInstruct) highlights the effectiveness of this configuration. Notably, we achieve these results without requiring hyperparameter tuning for specific datasets. This demonstrates that the hyperparameters of our method are robust and not sensitive to model size, dataset size, or task type, enabling excellent generalization across various scenarios.
>
> (b). During the rebuttal period, we conducted an additional ablation study for corset selection to validate our claim regarding the number of checkpoints. This study was performed on two large-scale scenarios: vision-language pre-training (VLP) using CLIP-ViT-Base and supervised fine-tuning (SFT) of a large language model on the OpenMathInstruct dataset.
>
>
> First, we present the VLP results at a selection rate of 10\% in the table below. The results indicate that as $m$ increases, both performance and stability improve. However, this improvement plateaus once $m$ exceeds 5.
>
> The choice of m|Zero-shot classification on ImageNet-1K|Linear Prob on ImageNet-1K|I2T Retrieval|T2I Retrieval
> ---|---|---|---|---
> 3|24.3$\pm$1.25|33.2$\pm$0.42|20.7$\pm$1.14|46.5$\pm$1.21
> 5|25.8$\pm$0.43|34.4$\pm$0.10|23.9$\pm$0.82|48.3$\pm$0.79
> 20|26.1$\pm$0.21|34.6$\pm$0.11|24.4$\pm$0.65|48.2$\pm$0.31
>
> Next, we present the coreset experimental results for supervised finetuning Llama-3-8B-Instruct at a selection rate of 10\% on the OpenMathInstruct dataset. This dataset comprises 1.8 million problem-solution pairs designed for math instruction. As detailed in Appendix D.4, the results reveal a similar trend: increasing $m$ enhances performance and stability, but the improvement saturates when $m$ exceeds 5.
>
> Further, for the sampling method, on the coreset experiments of ResNet-50 and ImageNet-1K at a selection rate of 20\%,  we validate that uniform random sampling is a good choice. Here, we experimented with fixed-step selection (mode-0) and uniform random sampling (mode-1), both of which achieved similar performance. We also compare the other two selection modes, namely, mode-2 selecting checkpoints from the first 5 epochs; and mode-3 selecting checkpoints from the last 5 epochs. These two modes perform inferiorly. Consequently, we adopted uniform random sampling in our experiments.
>
> Mode|Accuracy (top-1)
> ---|---
> mode-0|61.6
> mode-1|61.7$\pm0.12$
> mode-2|54.3
> mode-3|60.2
>
> In summary, these findings further validate that the hyperparameters optimized on the CIFAR-10 dataset with a ResNet-18 backbone generalize well across different datasets and models. This demonstrates that hyperparameter tuning is unnecessary for new tasks and models, ensuring broad applicability with high efficiency.

---

> ### Author Response · Authors · 2024-11-23
> **Response to Reviewer 7tcF (Part-2)**
>
> **Weakness.1-2. How the randomness of the selection affects the performance**
>
> Based on the above experiments and the ablation results presented in Figures 3(a) and 3(b) in Sec. 5.4, the small standard deviation (less than 0.05 in the approximation accuracy test) across multiple experimental runs on CIFAR-10 suggests that the randomness introduced by random sampling becomes negligible when $m\geq5$. We attribute this to the fact that uniform sampling ensures the selected checkpoints are evenly distributed across different training stages, capturing diverse model behaviors throughout learning.
>
> This observation also highlights a key guideline for checkpoint selection: sample evenly to ensure that time steps span across all training stages, providing comprehensive coverage of the model's evolution during learning.
>
>
> To validate this selection strategy, we conducted additional experiments on the coreset experiments of ResNet-50 and ImageNet-1K at a selection rate of 20\%, comparing fixed-step selection (Mode-0) and uniform random sampling (Mode-1). As shown in the table below both methods achieved comparable performance. We also evaluated two alternative selection modes: Mode-2, which selects checkpoints from the first 5 epochs, and Mode-3, which selects checkpoints from the last 5 epochs. These two modes performed noticeably worse.
>
> Notably, for Mode-1, the performance variance was within 0.12\% across 5 runs, further confirming that the randomness introduced by sampling is negligible as long as the sampling spans different learning stages.
>
>
> Mode|Accuracy (top-1)
> ---|---
> mode-0|61.6
> mode-1|61.7$\pm0.12$
> mode-2|54.3
> mode-3|60.2
>
>
>
>
> **Weakness.2-1: Discussion and comparison with [1,2,3,4,5]**
>
> We sincerely thank the reviewers for providing these valuable references. EK-FAC [1], TRAK [2], and Arnoldi-IF [3] leverage Kronecker factorization, kernel estimation techniques, and Arnoldi acceleration, respectively, to improve the scalability of influence estimations (IF). Despite these advancements, [1] and [2] rely on the convexity assumption of loss functions, which is often violated in practical scenarios. While Arnoldi-IF [3] improves scalability through the Arnoldi method, however, it still requires the assumption of the network's convexity. OGI [5] connects the identification of harmful training samples with outlier detection by applying influence functions in the gradient space. Both [4] and [5] perform well in handling noisy and outlier data.
>
> In response to the reviewers’ suggestions, we have conducted additional experiments comparing our approach with [1], [2], and [3] on the coreset selection task, as well as with [4] and [5] on the data cleaning task.
>
> (a)  First, we present the coreset experiment results. Specifically, we evaluated Llama-3-8B-SFT at a selection rate of 10\% using the OpenMathInstruct dataset, which includes 1.8 million problem-solution pairs. For additional details and discussions, please refer to Appendix D.4. The results demonstrate that Diff-In outperforms all other methods, highlighting its effectiveness in measuring sample importance for large datasets and models.
>
> Method|8-shot (CoT) on GSM8K
> ---|---
> Llama-3-8B-Instruct|77.2
> SFT on 1.8M data|85.3
> SR=30\% by EK-FAC[1]|80.3
> SR=30\% by TRAK[2]|82.7
> SR=30\% by Arnoldi[3]|80.4
> SR=30\% by Diff-In|84.4
>
> (b) Next, we conducted data-cleaning experiments on the GSM8K dataset, comparing Diff-In with IP [4] and OGI [5]. The experimental setup aligns with the description in Sec.5.1. In these experiments, Diff-In consistently achieved significantly superior performance, further demonstrating its effectiveness.
>
> Method|Precision on GSM8K
> ---|---
> SR=20\% by IF|67.2
> SR=20\% by IP[4]|74.4
> SR=20\% by OGI[5]|81.2
> SR=20\% by Diff-In |86.1
>
> **Weakness.2-2. Data cleaning using validation influence.**
>
> The table below presents the results of the data-cleaning task after replacing the self-influence estimated by Diff-In with validation influence. A slight drop in performance was observed, suggesting that self-influence may be a more effective method for identifying incorrect or outlier data. Additionally, self-influence is computationally more efficient, as it requires only the sample itself to calculate influence, rather than relying on a separate validation dataset.
>
> Method|Precision on GSM8K
> ---|---
> SR=20\% by Diff-In (Self influence)|86.1
> SR=20\% by Diff-In (Validation influence)|85.8
>
> A high self-influence value indicates that the model lacks sufficient training samples to make predictions consistent with the labeled information. This often serves as a reliable signal for detecting incorrect data or outliers.

---

> ### Author Response · Authors · 2024-11-23
> **Response to Reviewer 7tcF (Part-3)**
>
> **Weakness.2-3. For experiments on large models with large datasets, the selection of m should be considered.**
>
> Please refer to our responses to 1-1 and 1-2 for details on why selecting $m=5$ is generalizable across different model sizes and tasks.
>
> Here, we present additional experiments in LLM SFT scenarios to evaluate Diff-In's performance and runtime efficiency with varying values of $m$. More details are provided in Appendix D.4. Diff-In consistently delivers the best performance. As $m$ increases, both performance and stability improve, with diminishing returns once $\geq 5$. These findings align with the ablation study in the main paper and our responses to 1-1 and 1-2.
>
> We observed that with 5 checkpoints, Diff-In incurred only about 2 additional GPU hours compared to TracIn while improving performance by over 3\%. This checkpoint configuration is robust across different settings and does not require re-validation on new datasets and tasks.
>
> Method|8-shot (CoT) on GSM8K|Selection time cost
> ---|---|---
> Llama-3-8B-Instruct|77.2|0
> SFT on 1.8M data|85.3|0
> SR=30\% by TracIn with m=5|81.3$\pm$0.23|26.3 GPU-hours
> SR=30\% by Diff-In with m=3|82.1$\pm$1.03|17.4 GPU-hours
> SR=30\% by Diff-In with m=5|84.4$\pm$0.39|28.7 GPU-hours
> SR=30\% by Diff-In with m=10|84.6$\pm$0.11|56.8 GPU-hours
>
>
>
>
>
> **Weakness.3. Concerns about the experiment and title on the data deletion subsection**
>
> Thank you for the great suggestion! We’ve added clarifications in Section 5.2 to emphasize that our focus is on unlearning data points without retraining. Additionally, we included two unlearning baselines, SSSE [a] and PR [b], from [7] for comparison. Despite our approach being broadly applicable to various data-centric tasks, it still outperforms methods specifically designed for the unlearning task.
>
> Method|CIFAR-100|Tiny-ImageNet
> ---|---|---
> Diff-In|73.5|42.9
> SSSE [a]|72.2|41.5
> PR [b]|68.4| 39.2
>
> [a]. Efficiently Erasing Samples from Trained Machine Learning Models. NeurIPS 2021 Workshop.
>
> [b]. Machine Unlearning Method Based On Projection Residual. DSSA 2022.
>
>
>
>
>
> **Weakness.4. Some related works need to be discussed [8,9,10,11,12,13,14].**
>
> Thanks! We have already added them in the related work section in the revision.
>
>
>
> **Weakness.5. The reviewer pointed out some typos.**
>
> Thanks! We have revised them all!
>
>
> ---
>
> [1]. Studying Large Language Model Generalization with Influence Functions. Arxiv 2308.03296.
>
> [2]. TRAK: Attributing Model Behavior at Scale. ICML 2023.
>
> [3]. Scaling Up Influence Functions. AAAI 2022.
>
> [4]. Revisit, Extend, and Enhance Hessian-Free Influence Functions. Arxiv 2405.17490
>
> [5]. Outlier Gradient Analysis: Efficiently Improving Deep Learning Model Performance via Hessian-Free Influence Functions. Arxiv 2405.03869.
>
> [6]. DataInf: Efficiently Estimating Data Influence in LoRA-tuned LLMs and Diffusion Models. ICLR 2024.
>
> [7]. A Survey of Machine Unlearning. Arxiv 2209.02299.
>
> [8]. Datamodels: Predicting Predictions from Training Data. ICML 2022.
>
> [9]. What is Your Data Worth to GPT? LLM-Scale Data Valuation with Influence Functions. ArXiv 2405.13954
>
> [10]. "What Data Benefits My Classifier?" Enhancing Model Performance and Interpretability through Influence-Based Data Selection. ICLR 2024.
>
> [11]. HYDRA: Hypergradient Data Relevance Analysis for Interpreting Deep Neural Networks. AAAI 2021.
>
> [12]. Achieving Fairness at No Utility Cost via Data Reweighing with Influence. ICML 2022.
>
> [13]. Revisiting inverse Hessian vector products for calculating influence functions. Arxiv 2409.17357.

---

> > ### Comment · Reviewer_7tcF · 2024-11-23
> >
> > Dear authors, thank you for your detailed response and additional experiments. The new results resolve my concerns and I have increased my score accordingly. Please also ensure that all these new experiments, results, and discussion are added to the paper revision, as I did not find everything in the current version.

---

> ### Author Response · Authors · 2024-11-25
> **A Grateful Response to Reviewer 7tcF**
>
> We appreciate your recognition and your reply!
>
>
> We have added the experimental content of this Rebuttal and the discussion content of general interest to the article, and the specific updates are as follows. Please let us know if you have any further interest or concerns.
>
> Specifically, we have made the following changes:
>
>     1. Fixed typos (main paper).
>
>     2. Added new references [1-13] in the related work section (appendix, Sec.C).
>
>     3. Added clarification of Data deletion and new baseline [14, 15] (main paper).
>
>     4. Added speed and performance comparison with SOTA methods on experiments for Llama-3-8B on OpenMathInstruct (appendix, Sec.D.4.1 and Table 11).
>
>     5. Added further ablation study on the choice of the number of selected checkpoints (appendix, Sec.D.4.2 and Table.11 and Table 13).
>
>     6. Added further performance comparison with IP and OGI on the cleaning experiments for GSM8K  (appendix, Sec.D.4.1 and Table.12).
>
>     7. Added guidelines for checkpoint selection (appendix, Sec.D.4.3).
>
>     8. Added experiments and discussion about the growth of the bound (appendix, Sec.D.5.1).
>
>     9. Added discussion about the assumption of the bound (appendix, Sec.D.5.2).
>
>     10. Moved the discussion of limitations to the appendix.
>
> **Reference:**
>
>     [1]. Studying Large Language Model Generalization with Influence Functions. Arxiv 2308.03296.
>
>     [2]. TRAK: Attributing Model Behavior at Scale. ICML 2023.
>
>     [3]. Scaling Up Influence Functions. AAAI 2022.
>
>     [4]. Revisit, Extend, and Enhance Hessian-Free Influence Functions. Arxiv 2405.17490
>
>     [5]. Outlier Gradient Analysis: Efficiently Improving Deep Learning Model Performance via Hessian-Free Influence Functions. Arxiv 2405.03869.
>
>     [6]. DataInf: Efficiently Estimating Data Influence in LoRA-tuned LLMs and Diffusion Models. ICLR 2024.
>
>     [7]. A Survey of Machine Unlearning. Arxiv 2209.02299.
>
>     [8]. Datamodels: Predicting Predictions from Training Data. ICML 2022.
>
>     [9]. What is Your Data Worth to GPT? LLM-Scale Data Valuation with Influence Functions. ArXiv 2405.13954
>
>     [10]. "What Data Benefits My Classifier?" Enhancing Model Performance and Interpretability through Influence-Based Data Selection. ICLR 2024.
>
>     [11]. HYDRA: Hypergradient Data Relevance Analysis for Interpreting Deep Neural Networks. AAAI 2021.
>
>     [12]. Achieving Fairness at No Utility Cost via Data Reweighing with Influence. ICML 2022.
>
>     [13]. Revisiting inverse Hessian vector products for calculating influence functions. Arxiv 2409.17357
>
>     [14]. Efficiently Erasing Samples from Trained Machine Learning Models. NeurIPS 2021 Workshop.
>
>     [15]. Machine Unlearning Method Based On Projection Residual. DSSA 2022.
>
> Thank you very much again!

---

### Official Review · Reviewer_yitw · 2024-11-04

**Soundness:** 3
**Presentation:** 3
**Contribution:** 3
**Rating:** 6
**Confidence:** 4

**Summary:**

This paper proposes Diff-In to estimate the data influence by accumulating differences between consecutive training steps. Diff-In approximates data influence as temporal differences with second-order methods. However, its computational cost is comparable with a first-order method.

I am not an expert in data influence. I feel my assessment of the paper's novelty may not be accurate. However, I acknowledge the theoretical and empirical results presented in the paper, and think that this is a solid paper.

**Strengths:**

1. Novel Theoretical Framework: Provides rigorous mathematical formulation with error bounds for influence estimation without requiring model convexity.

2. Computational Efficiency: Achieves complexity comparable to first-order methods despite using second-order approximations through efficient Hessian-gradient product calculations.

3. Strong Empirical Results: Demonstrates consistent superior performance across data cleaning (9% improvement), data deletion (2% improvement), and coreset selection tasks on multiple datasets.

**Weaknesses:**

1. Checkpoint Dependency: Implementation relies heavily on saved checkpoints during training, with unclear guidelines on optimal checkpoint selection.

2. Limited Generalizability: Currently focused on sample-level influence, lacking broader application to model hyperparameters.

3. Theoretical Assumptions: Analysis assumes Lipschitz continuous gradients and bounded gradient norms, with limited discussion of assumption violations.

**Questions:**

Please refer to the weaknesses.

---

> ### Author Response · Authors · 2024-11-23
> **Response to Reviewer yitw**
>
> Thank you for your time and effort during the review process, as well as for your recognition and thoughtful suggestions regarding our paper. Your feedback is highly valuable and has significantly contributed to improving the quality of our paper.
>
> ---
>
> **Weakness.1 The implementation relies heavily on saved checkpoints during training, with unclear guidelines on optimal checkpoint selection.**
>
> Thanks!
>
> (a) The checkpoint selection process involves determining the number of checkpoints, $m$, and the sampling method. These parameters were optimized through ablation experiments conducted on the CIFAR-10 dataset using the ResNet-18 backbone, as detailed in Sec. 5.4.
>
> (b) For the selection of $m$, as shown in Figures 3(a) and 3(b), increasing $m$ improves approximation precision and reduces variance but also increases computation time. To balance computational cost and precision, we set $m=5$ as performance tends to grow slowly after this point while maintaining high efficiency. During the rebuttal, we have further conducted coreset selection experiments on the OpenMathInstruct dataset for fine-tuning Llama-3-8B (see Appendix.D.4 for details). OpenMathInstruct is a large math instruction tuning dataset with 1.8M problem-solution. Here are the results. We can observe a similar trend, that is, the increase of m will improve the performance and stability, but this improvement will be saturated at m greater than 5.
>
> Method|8-shot (CoT) on GSM8K
> ---|---
> Llama-3-8B-Instruct|77.2
> SFT on 1.8M data|85.3
> SR=30\% by Diff-In with m=3|82.1$\pm$1.03
> SR=30\% by Diff-In with m=5|84.4$\pm$0.39
> SR=30\% by Diff-In with m=10|84.6$\pm$0.11
>
>
> (c) For the sampling guideline, on the coreset experiments of ResNet-50 and ImageNet-1K at a selection rate of 20\%, we experimented with fixed-step selection (mode-0) and uniform random sampling (mode-1), both of which achieved similar performance. We also compare the other two selection modes, namely, mode-2 selecting checkpoints from the first 5 epochs; and mode-3 selecting checkpoints from the last 5 epochs. These two modes perform inferiorly. Consequently, we adopted uniform random sampling in our experiments.
>
> Mode|Accuracy (top-1)
> ---|---
> mode-0|61.6
> mode-1|61.7$\pm0.12$
> mode-2|54.3
> mode-3|60.2
>
> (d) In all experiments presented in the main paper, we applied the above checkpoint selection strategy. The promising results are shown in Tables 1, 2, 3, and 4 across various tasks and datasets, such as coreset for ImageNet-1K and CC12M, and data clean for GSM8K, demonstrating the effectiveness and generalizability of this strategy to a wide range of datasets.
>
> **Weakness.2 Limited Generalizability: focusing on sample-influence, lacking broader application to hyperparameters.**
>
> Thank you for the thoughtful suggestion! Existing methods for influence estimation, including ours, rely on calculating derivatives, which require the function to be differentiable. As a result, the non-differentiable nature of the loss function with respect to model hyperparameters makes it difficult to directly apply such methods to estimate the influence of hyperparameters on model learning.
>
> In the future, leveraging estimation methods (e.g., straight-through estimation) to estimate gradients could be a promising solution for incorporating hyperparameter influence into the existing framework, which is beyond the scope of this paper, focusing on data influence.
> We consider it an exciting direction for future exploration.
>
>
>
> **Weakness.3 Theoretical Assumptions: Analysis assumes Lipschitz continuous gradients and bounded gradient norms, with limited discussion of assumption violations.**
>
> Thank you for your comments!
>
> Lipschitz continuity and gradient norms are commonly used to characterize the smoothness of a neural network's loss landscape. Modern deep learning models incorporate techniques such as normalization layers (e.g., batch normalization and layer normalization) and shortcut connections to enhance smoothness and continuity [6,7], facilitating optimization. These techniques make the assumption of smoothness generally valid in practice. This stands in contrast to many studies [3,4] that rely heavily on the convexity of neural networks. Our assumption, by comparison, is much easier to satisfy in real-world scenarios.
>
> If the conditions for $\ell$-Lipschitz continuous gradients are not met, the gradient norms $g$, the values of $g$ and $\ell$ can become very large. In such cases, the error bound derived from these parameters may lose its practical relevance. However, this does not necessarily imply that the algorithm will fail in practice.
>
> We include the discussion in Appendix.D.5.

---

> ### Author Response · Authors · 2024-11-23
> **Reference**
>
> [1]. Estimating training data influence by tracing gradient descent. NeurIPS-2020.
>
> [2]. Data cleansing for models trained with SGD. NeurIPS-2019.
>
> [3]. Understanding black-box predictions via influence functions. ICML-2017.
>
> [4]. Datainf: Efficiently estimating data influence in lora-tuned llms and diffusion models. ICLR 2024.
>
> [5]. Learning transferable visual models from natural language supervision. ICML-2021.
>
> [6]. How Does Batch Normalization Help Optimization? NeurIPS-2018.
>
> [7]. Visualizing the Loss Landscape of Neural Nets. NeurIPS-2018.

---

> ### Comment · Reviewer_yitw · 2024-11-25
>
> Thank the authors for the detailed responses! I read the responses and the updated PDF. My concerns have been mostly addressed.

---

### Author Response · Authors · 2024-11-23
**General Response to ACs and Reviewers**

Dear Reviewers and ACs:

We sincerely appreciate your constructive comments and insightful reviews, which have significantly contributed to enhancing our work. We have thoroughly considered all your suggestions and made substantial revisions to our previous draft, with the main changes highlighted in blue.

Specifically, we have made the following changes:

    1. Fixed typos (main paper).

    2. Added new references [1-13] in the related work section (appendix, Sec.C).

    3. Added clarification of Data deletion and new baseline [14, 15] (main paper).

    4. Added speed and performance comparison with SOTA methods on experiments for Llama-3-8B on OpenMathInstruct (appendix, Sec.D.4.1 and Table 11).

    5. Added further ablation study on the choice of the number of selected checkpoints (appendix, Sec.D.4.2 and Table.11 and Table 13).

    6. Added further performance comparison with IP and OGI on the cleaning experiments for GSM8K  (appendix, Sec.D.4.1 and Table.12).

    7. Added guidelines for checkpoint selection (appendix, Sec.D.4.3).

    8. Added experiments and discussion about the growth of the bound (appendix, Sec.D.5.1).

    9. Added discussion about the assumption of the bound (appendix, Sec.D.5.2).

    10. Moved the discussion of limitations to the appendix.

**Reference:**

    [1]. Studying Large Language Model Generalization with Influence Functions. Arxiv 2308.03296.

    [2]. TRAK: Attributing Model Behavior at Scale. ICML 2023.

    [3]. Scaling Up Influence Functions. AAAI 2022.

    [4]. Revisit, Extend, and Enhance Hessian-Free Influence Functions. Arxiv 2405.17490

    [5]. Outlier Gradient Analysis: Efficiently Improving Deep Learning Model Performance via Hessian-Free Influence Functions. Arxiv 2405.03869.

    [6]. DataInf: Efficiently Estimating Data Influence in LoRA-tuned LLMs and Diffusion Models. ICLR 2024.

    [7]. A Survey of Machine Unlearning. Arxiv 2209.02299.

    [8]. Datamodels: Predicting Predictions from Training Data. ICML 2022.

    [9]. What is Your Data Worth to GPT? LLM-Scale Data Valuation with Influence Functions. ArXiv 2405.13954

    [10]. "What Data Benefits My Classifier?" Enhancing Model Performance and Interpretability through Influence-Based Data Selection. ICLR 2024.

    [11]. HYDRA: Hypergradient Data Relevance Analysis for Interpreting Deep Neural Networks. AAAI 2021.

    [12]. Achieving Fairness at No Utility Cost via Data Reweighing with Influence. ICML 2022.

    [13]. Revisiting inverse Hessian vector products for calculating influence functions. Arxiv 2409.17357

    [14]. Efficiently Erasing Samples from Trained Machine Learning Models. NeurIPS 2021 Workshop.

    [15]. Machine Unlearning Method Based On Projection Residual. DSSA 2022.

Thank you very much again!

Best regards,
Authors of Paper1257

---

### Meta-Review · Area_Chair_36q5 · 2024-12-13

**Metareview:**

I have read all the materials of this paper including the manuscript, appendix, comments, and response. Based on collected information from all reviewers and my personal judgment, I can make the recommendation on this paper, reject. No objection from reviewers who participated in the internal discussion was raised against the reject recommendation.

**Research Question**

This paper considers the sample influence estimation problem.

**Challenge Analysis**

The authors argue that the current methods suffer from the convex assumption and Hessian-free method limit its accuracy.

**Philosophy**

The authors aim to present the sample influence as the cumulative sum between successive training steps, without replying on convexity assumption. Note that TracIn has a similar philosophy (See Eq.(52) in [1]) and this is not a weakness from my eyes.

**Solution**

The authors propose Diff-In, a second-order method for the cumulative sum between successive training steps. To reduce the computational cost, the authors estimate the product of Hessian and gradient using finite differences of first-order gradients.  I believe such cumulation can alleviate the convexity assumption, however, I do not see any novelty in computational saving, where the authors employ mature techniques. Checkpoints are well studies in this paper, especially when few checkpoints are selected, it is uncertain whether the claimed unrelaxing assumption holds.

**Theoretical Analysis**

One reviewer pointed out that the theoretical upper limit of the Diff-In estimation error grows polynomially with the number of training steps. Respectively and personally, I do not think this theorem is very helpful in practice, or brings in computational saving, or sheds any insights in parameter selection. Not every paper needs this part.

**Experiments**

I have big concerns on this part.

1.	The experiment setup needs more details. The current version including appendix does have the parameters used in the proposed method. For example, the gradient is the complete one or just the last layer.

2.	The competitive methods in Table 2 are not consistent with Table 1. At least, influence-function based methods should be added in Table 2.

3.	The datasets used in different tasks are also different. Why?

4.	I have concerns about the running time in Figure 3(b). It seems that Diff-In has similar execution time with TracIn. It seems that calculating the batch Hessian does not take much time. Why?

**Other issues**

1.	Figure 1 plays a good motivation on the inaccuracy of existing influence function-based methods. However, the results are questionable. (i) Since the performance of deep models is affected by random seeds, the results are not deterministic even with the whole training set. Moreover, removing a single training sample may not have a significant impact, especially when considering the inherent randomness in the training process. Based on my personal experience on using ResNet on CIFAR-10 and comparing performance with the complete training set versus with one sample removed, after running these experiments multiple times, the average performance between the two settings did not show statistical significance. (ii) The definition of “30 most influential data points” is unclear. The most influential is based on which algorithm? If it is based on the proposed method, the results might be biased.

[1] Training Data Influence Analysis and Estimation: A Survey.

**Additional Comments On Reviewer Discussion:**

No objection from reviewers who participated in the internal discussion was raised against the reject recommendation.

Issues with checkpointing and hyper-parameter tuning were brought up by reviewers and the authors addressed this in the rebuttal. Although Review 7tcf was initially satisfied with the promised new results, the final revised version of the paper does not satisfactorily provide the experimental evidence promised. In subsequent discussions with Review 7tcf, we found that the final results did not sufficiently support the arguments that initially led the reviewer to raise their score.

---

### Decision · Program_Chairs · 2025-01-22

Reject